



# 1 A transition in atmospheric emissions of particles and gases from

# 2 on-road heavy-duty trucks

Liyuan Zhou[1], Åsa M. Hallquist[2*], Mattias Hallquist[3], Christian M. Salvador[3], Samuel M. Gaita[3], Åke
Sjödin[2], Martin Jerksjö[2], Håkan Salberg[2], Ingvar Wängberg[2], Johan Mellqvist[4], Qianyun Liu[1], Berto P.
Lee[1], Chak K. Chan[1*]
[1]School of Energy and Environment, City University of Hong Kong, Hong Kong, China
[2]IVL Swedish Environmental Research Institute, Gothenburg, Sweden
[3]Department of Chemistry and Molecular Biology, University of Gothenburg, Gothenburg, Sweden
[4] Earth and Space Sciences, Chalmers University of Technology, Gothenburg, Sweden
*Correspondence to*: Åsa M. Hallquist (asa.hallquist@ivl.se), Chak K. Chan (Chak.K.Chan@cityu.edu.hk)
**Abstract.** The transition in extent and characteristics of atmospheric emissions caused by the modernisation of the heavy-duty
on-road fleet were studied utilising roadside measurements. Emissions of particle number (PN), particle mass (PM), black
carbon (BC), nitrogen oxides ($NO_x$), carbon monoxide (CO), hydrocarbon (HC), particle size distributions and particle
volatility were measured from 556 individual heavy-duty trucks (HDTs). Substantial reductions in PM, BC, $NO_x$, CO and to
a lesser extent PN were observed from Euro III to Euro VI HDTs by 99%, 98%, 93% and 57% for the average emissions
factors of PM, BC, $NO_x$, and CO respectively. Despite significant total reductions in $NO_x$ emissions, the fraction of $NO_2$ in the
$NO_x$ emissions increased continuously from Euro IV to Euro VI HDTs. Larger data scattering was evident for PN emissions
in comparison to solid particle number (SPN) for Euro VI HDTs, indicating a highly variable fraction of volatile particle
components. Particle size distributions of Euro III to EEV HDTs were bimodal, whereas those of Euro VI HDTs were
nucleation mode dominated. High emitters disproportionately contributed to a large fraction of the total emissions with the
highest-emitting 10% of HDTs in each pollutant category being responsible for 65% of total PM, 70% of total PN and 44% of
total $NO_x$ emissions, respectively. Euro VI HDTs, which accounted for 53% of total kilometres driven by Swedish HDTs, were
estimated to only contribute to 2%, 6%, 12% and 47% of PM, BC, $NO_x$, and PN emissions. A shift to a Euro VI HDTs dominant
fleet would promote a transition of atmospheric emissions towards low PM, BC, $NO_x$, and CO levels. Nonetheless, reducing
PN, SPN, and $NO_2$ emissions from Euro VI HDTs is still important to improve air quality in urban environments.

## 26 1 Introduction

Vehicular emissions contribute significantly to gaseous and particle pollutants in the urban atmosphere and description of their
extent and characteristics are key input components for urban air quality modelling. As technology and traffic demands change,
so do the characteristics of the emissions. In Europe, the introduction of new legislation, especially Euro VI, has aimed to



reduce emissions of many pollutants. Legislation exists for particles (mass and number) and selected gases, however, there are
also many components of the emissions that are not directly regulated but are potentially detrimental to human health. The
most notable example of a non-regulated pollutant is the abundance of ultrafine particles (UFP) (Campagnolo et al., 2019),
defined as particles with a diameter less than 100 nm (Zhu et al., 2002). UFPs can cause lung disease, an increase of blood
coagulability and cardiovascular disease and related mortality (Du et al., 2016). In the most recent Euro class, this has partly
been covered by introducing a limit on the solid particle number (SPN) while volatile particles and particles less than 23 nm
are not considered. Furthermore, the legislation has mainly been based on test cycles performed before introducing a new
engine into the market but only recently also off-cycle and in-service conformity testing has been introduced, hence the actual
performance in real-traffic is less constrained, where driving pattern, maintenance, and age of engine will vary. Here real-
traffic studies may capture variability between vehicles and also put the effect of new legislation and parallel phase-out of
older technology into perspective for the abatement of urban air pollution.
Heavy-duty vehicles (HDVs) usually account for a smaller number fraction of on-road vehicles than light-duty vehicles
but they tend to contribute to a disproportionately high fraction of mobile source particulate matter emissions (Gertler, 2005;
Cui et al., 2017). Emissions from HDVs, often diesel, are significantly affected by the engine type, exhaust after-treatment
system (ATS), and driving conditions. The main purpose of ATS is the reduction of particulate and gaseous pollutants. Diesel
Oxidation Catalysts (DOC) are used for reducing hydrocarbon emissions, selective catalytic reduction (SCR) systems or
exhaust gas recirculation (EGR) are employed to mitigate $NO_x$ emissions, and diesel particulate filters (DPF) can reduce
particulate matter mass emissions. The use of ATS can, however, bring undesired side effects. For example, conversion of $SO_2$
to $SO_3$ and increased gaseous sulfuric acid formation have been reported from DOCs (Arnold et al., 2012). DPFs potentially
enhance the formation of UFP (Herner et al., 2011; Preble et al., 2017). Retrofitted DPF can slightly reduce the $NO_x$ emissions
but significantly increase the direct emission of $NO_2$ by as much as a factor of 8 (Smith et al., 2019). Failure of the temperature-
dependent SCR in eliminating the excess $NO_2$ leads to an elevated $NO_2$ to $NO_x$ ratio (Herner et al., 2009; Bishop et al., 2010;
He et al., 2015).
The Euro standard regulates vehicle emission limits in Europe. The Euro III standard was established in 1999, and more
stringent Euro IV and Euro V standards were implemented in 2005 and 2008, respectively. The Enhanced Environmentally
Friendly Vehicle (EEV) is a voluntary environmental standard which lies between the levels of Euro V and Euro VI. The
currently enforced Euro VI standard has been implemented since 2013-2014 and introduced SPN limits into the regulation for
the first time. Generally, newer engines are expected to perform better in controlling pollutant emissions. Guo et al. (2014)
reported that Euro V diesel buses performed better than Euro IV and Euro III diesel buses in the emissions of all the pollutants,
except for the generation of more nucleation mode particles. The latest 2018 European Environmental Agency (EEA) report
confirms an overall improvement in the European air quality, while the road transport sector remains one of the major
contributors to pollutant emissions and the largest contributor to the total $NO_x$ emission (Grigoratos et al., 2019; EEA, 2018).
A recent on-board sensor-based study pointed out that HDVs emitted more than three times the $NO_x$ certification standard
during real-world hot-driving and idling operations (Tan et al., 2019). Published data regarding particle and gaseous pollutant



emissions from real-world on-road Euro VI heavy-duty vehicles are scarce and often limited by the small sample size (Giechaskiel et al., 2018; Grigoratos et al., 2019; Moody and Tate, 2017). Remote sensing sampling can measure a large sample size of vehicles but are usually restricted to gaseous pollutant emissions (Burgard and Provinsal, 2009; Burgard et al., 2006; Carslaw et al., 2011). From an air quality perspective, the particle emissions are crucial. The complexity and dynamics of atmospheric particles require detailed information of its emission for atmospheric modelling and for descriptions of their health impacts. For example, particle size is important to determine the effects on respiratory deposition in humans (Manigrasso et al., 2017; Lv et al., 2016).

Diesel exhaust particles are a complex mixture of numerous semi-volatile and non-volatile species, and the semi-volatile compounds will experience gas-to-particle partitioning in the atmosphere (Robinson et al., 2007; Donahue et al., 2006). Biswas et al. (2009) reported that the semi-volatile fraction in HDV emission is more oxidative than the refractory particles, which may change the redox state in cells and cause oxidative stress. Semi-volatile organic compounds, such as PAHs and their derivatives may possess genotoxic and carcinogenic characteristics (Bocchi et al., 2016; Vojtisek-Lom et al., 2015). Giechaskiel et al. (2009) suggested using the volatile mass fraction as a metric to assess health effects as the volatile mass dissolves in the lung fluid and thereby interacts with epithelial cells. Deploying a Volatility Tandem Differential Mobility Analyzer in suburban Guangzhou, China, Cheung et al. (2016) found that 57–71 % of ambient particles between 40 and 300 nm contain volatile components. Furthermore, the evaporated semi-volatile compounds from the particle phase can be further oxidized to form secondary organic aerosols (SOA) (Hallquist et al., 2009; Gentner et al., 2017). To better quantify the health effects and global and regional contributions of road traffic to the total particle burden in the atmosphere, information on the volatility properties of vehicle particulate emissions is needed.

Different approaches have been applied to study the emissions from HDTs (Franco et al., 2013). Chassis dynamometer tests provide relatively comprehensive emission characteristics of individual vehicles (Jiang et al., 2018; Chen et al., 2018; Thiruvengadam et al., 2015), but the artificial driving cycles make it difficult to simulate the full range of real-world driving conditions. Portable emission measurement systems (PEMS) (Grigoratos et al., 2019; Pirjola et al., 2017) and plume chasing studies (Lau et al., 2015; Pirjola et al., 2016) have been conducted in real-world environments but are often limited by small sample sizes. Tunnel studies (Li et al., 2018) measure the average emission of all vehicles passing through the tunnel without specific emission information of vehicle types. Roadside measurements, as presented in this study, provide an opportunity to study real-world on-road traffic emissions on large sample sizes with individual vehicle information (e.g. Hallquist et al., 2013; Dallmann et al., 2012; Carslaw and Rhys-Tyler, 2013; Watne et al., 2018).

In this work, we measured the gaseous and particle emissions from 556 on-road individual HDTs and quantified changes in emissions and the potential transition in characteristics caused by the reduction achieved by the introduction of more stringent Euro standards. Particle size distributions and particle volatilities were investigated with respect to Euro class, and pollutant emission characteristics were studied with respect to year of registration. Cumulative pollutant distributions were established to demonstrate the importance of controlling high-emitters in reducing total emissions. The typical contribution of air pollution emissions from each Euro class HDTs was estimated based on total vehicle kilometres driven. Results of the





presented pollutant emission factors in our study will be useful for both emission models and emission inventories. A clear
transition of atmospheric pollutant emission trends was evident and can provide useful guidance for policies regarding the
regulation of existing fleets.

## 2 Methods

### 2.1 Field sampling site

Pollutant emissions from HDTs were measured at a roadside location in Gothenburg, Sweden (Fig. 1). The HDTs passed the
sampling location with an average speed of 27 km h$^{-1}$ and acceleration of 0.7 km h$^{-1}$ s$^{-1}$ on a slight uphill slope (~2°).

### 2.2 Air sampling

The sampling of the emissions was conducted in line with Hallquist et al. (2013), i.e. extractive sampling of passing HDT
plumes. Air was continuously drawn through a flexible copper tube to the instruments inside a container. A similar
experimental set-up was previously applied to on-road bus emission measurements (Liu et al., 2019). Particles were measured
by an EEPS (Engine Exhaust Particle Sizer, Model 3090 TSI Inc.) in the size range of 5.6-560 nm with high time resolution
(10 Hz) while total particle number was measured by a butanol-based condensation particle counter (CPC Model 3775 TSI
Inc., 50 cut-off diameter 4 nm). Particle numbers measured by the two instruments showed a good correlation ($R^2$=0.73)
(Fig. S1). A second EEPS measured the outflow of a TD (thermodenuder, Dekati, Inc.), enabling estimations of particle
volatility. The data were corrected for size-dependent losses in the TD. The temperature inside the TD heating zone was set to
250°C with a residence time of ~ 0.6 s, which is generally sufficient to evaporate nearly all the organics and sulphates from
the particles (Huffman et al., 2008). However, organics with extremely low volatility (organic saturation mass concentration,
C*< 10$^{-5}$µg m$^{-3}$ at 298 K) may still be retained even at this high temperature (Gkatzelis et al., 2016). Thus, in this study, we
define the 'non-volatile components' as particle components that remain after passing through the TD operating at 250°C.
Differences in counting efficiencies between the two EEPS were accounted for by size-dependent correction factors (typically
less than 10%), which were retrieved by simultaneous sampling of ammonium sulphate particles by both EEPS and direct
comparison of their measured size distributions (Fig. S2). BC and the mixture of BC and brown carbon (BrC) were measured
by an Aethalometer at 880 nm and 370 nm respectively (Model AE 33, Magee Scientific Inc.). The determination of particle
mass concentrations by the integrated particle size distribution (IPSD) method requires information on particle density. Particle
sphericity and unit density were assumed due to a lack of detailed knowledge about the chemical composition of the emitted
particles. Figure S3 shows that there is a good linear relationship between the BC mass measured by the Aethalometer and the
non-volatile particle mass measured by the EEPS but assuming sphericity and unit density the EEPS mass is lower, which
indicates a potential underestimation of the effective non-volatile particle density. There have been several studies on the
morphology and density of combustion generated particles and its detailed dependence on combustion and dilution condition



(e.g. Maricq and Ning, 2004; Ristimaki et al., 2007; Liu et al., 2009; Zheng et al., 2011; Quiros et al., 2015). However, to be
consistent, avoid assumptions and to compare with a majority of previously reported data, unity density was used for further
discussion and comparisons. $CO_2$ was measured by a non-dispersive infrared gas analyser (LI-840, LI-COR Inc.). $NO_x$ and
NO were measured simultaneously by two separate chemiluminescent analysers (Model 42i, Thermo Scientific Inc.), and the
$NO_2$ concentration was calculated from the difference between the $NO_x$ and NO concentrations. $SO_2$ was measured by a pulsed
fluorescence gas analyser (Model 43c, Thermo Scientific Inc.). A Remote Sensing Device (RSD) (OPUS Inspection Inc. )
used to measure the gaseous emission factors of CO, $NO_x$, and HC. Briefly, the instrument was set up with a transmitter and a
receiver on the same side of the truck passing lane and a reflector on the opposite side. Co-linear beams of IR and UV light
are emitted and cross-reflected through the plume and light attenuation related to respective pollutant concentrations are
measured. Gas pollutant concentrations were determined relative to $CO_2$ as measured by the RSD. $NO_x$ and NO measured by
the gas analysers and the RSD were in agreement ($R^2$=0.53 and 0.66 respectively, Figs. S4, S8a). The High-Resolution Time-
of-Flight Chemical Ionization Mass Spectrometer (HR-ToF-CIMS) shown in Fig. 1 was used to characterise the chemical
composition of organic and inorganic compounds in the gas and particle phase, emitted from the HDTs. However, the extensive
chemical characterisation is beyond the scope of this work and will be presented elsewhere.
All the instruments were operated at least at 1Hz of sampling frequency to capture rapidly changing concentrations during
the passage of a HDT. A camera at the roadside recorded the HDT plate numbers, which was used for identification and to
obtain engine Euro type information. A schematic of the experimental setup is given in Fig. 1 along with some examples of
typical temporal profiles of pollutant concentrations in the plumes from Euro III and Euro VI HDTs. In general, the duration
of a peak was around 5s, for $NO_x$ slightly longer, limiting measurements of high-frequency passages. Euro III HDTs typically
emitted a significant amount of PN, PM, $NO_x$, and non-volatile components (Fig. 1c). More than 95% of Euro III, Euro IV,
Euro V, and EEV HDTs had measurable particle emission signals. Significant differences in low particle and gaseous emissions
were evident for Euro VI HDTs (Fig. 1d and e).
**2.3 Data analysis**
The exact time of individual HDTs passing the sampling inlet was determined from the camera recordings and the associated
plume pollutant concentrations were integrated to calculate corresponding pollutant emission factors of individual HDTs as
described by Hallquist et al. (2013). Emissions of gases and particles from individual HDTs were normalized by the $CO_2$
concentration to compensate for different degrees of dilution during sampling (Janhäll and Hallquist (2005)). $CO_2$ peak
concentrations exceeding four times the standard deviation of the background signal were used as the base criterion for
successful plume capture. Peaks in $NO_x$, PN, PM, and BC concentrations concurrent with that of $CO_2$ signify the presence of
co-emitted pollutants in a HDT plume. Emission factors (EFs) of gaseous and particle emissions for individual HDTs can then
be expressed in units of amount of pollutant emitted per kg fuel burned based on the carbon balance method (Ban-Weiss et al.,
2009; Hak et al., 2009):



$$EF_{pollutant} = \frac{\int_{t_1}^{t_2}([pollutant]_t - [pollutant]_{t_1})dt}{\int_{t_1}^{t_2}([CO_2]_t - [CO_2]_{t_1})dt} \times EF_{CO_2} \,, \qquad (1)$$

where $EF_{pollutant}$ is the emission factor of the respective pollutant. The time interval of $t_1$ to $t_2$ represents the period when the instruments measured the concentration of an entire pollutant peak from an individual HDT (see Fig. 1c-e). $\int_{t_1}^{t_2}([pollutant]_t - [pollutant]_{t_1})dt$ and $\int_{t_1}^{t_2}([CO_2]_t - [CO_2]_{t_1})dt$ are the changes in concentration of a pollutant and $CO_2$ during this time interval. $EF_{CO2}$ of 3158 g (kg diesel fuel)$^{-1}$ was used as the emission factor of $CO_2$, assuming complete combustion and a carbon content of 86.1% as given in Edwards et al. (2014). Emission factors for plumes with pollutant concentrations lower than our set detection limit (four times the standard deviation of the pollutant background signal) were replaced by the minimum value among all recorded emission factors ($EF_{min}$) rather than being omitted to avoid overestimating emissions from low-emitting HDTs.

**3 Results and Discussion**

**3.1 Fleet compositions**

A total of 675 resolved plumes from 556 individual HDTs for the carriage of goods with weights exceeding 12 tonnes were identified. There were 330 Swedish HDTs with Euro type information, 46 Swedish HDTs from which Euro type information was not available, and 180 foreign licensed non-Swedish HDTs. Among the 330 Swedish trucks, Euro III, Euro IV, Euro V, EEV, and Euro VI HDTs accounted for 3%, 5%, 30%, 5%, and 57%, respectively (Fig. S5).

**3.2 Emissions variability**

Differences in operating and ambient conditions may lead to differences in pollutant emission factors for the same HDT. In this study, we utilized measurement data from 55 HDTs which passed the sampling location repeatedly, yielding a total of 137 plumes. The average pollutant emission factors of each HDT plotted against the individual plume measurements of the corresponding HDT are shown in Fig. S6. In general, the emission factors of PM, non-volatile PN, and $NO_x$ showed little variation ($R^2 \geq 0.77$) among multiple passages of the same HDT, however, higher variability was observed in the PN emissions. This is likely related to variations in the formation of nucleation mode particles from volatile compounds which is more sensitive to driving (Zheng et al., 2014) and dilution conditions. In the following discussion, for HDTs with multiple passages, the average pollutant emission factors of all the detected plumes were used for that individual HDT.

**3.3 Emissions factors (EFs) of particles and gaseous pollutants**

Figure 2 a and b show the box-and-whisker plots of PM and PN emission factors (EFs) for different Euro classes. Generally, both PM and PN emissions decreased with more stringent Euro emission standards, and especially for Euro VI where larger changes in emission characteristics were evident. Using Euro III HDTs (median $EF_{PM}$=586 mg (kg fuel)$^{-1}$) as a baseline, the



median EF$_{PM}$ for Euro IV, Euro V, EEV, and Euro VI HDTs have been reduced by 78.1%, 86.1%, 88.9%, and 99.8%
respectively. In particular, Euro VI HDTs has a median EF$_{PM}$ of only 1.4 mg (kg fuel)$^{-1}$ (Fig. 2a). While it is noted that Euro
III to Euro VI standard certifications are based on chassis dynamometer cycle measurements, the Euro VI regulations have
started to include additional off-cycle and in-service conformity testing. The Euro emission standard of transient testing for
heavy-duty engines gives emission limits as brake specific emission factors, as mass (g) or number (#) of a specific pollutant
per kWh. In order to enable a comparison with the Euro emission standard, the EFs in g (or #) per kg fuel were converted using
a brake specific fuel consumption (BSFC) of 231.5 g kWh$^{-1}$. This is the average value for the long haul and regional delivery
cycles of chassis dynamometer tests of a typical rigid Euro VI truck (Rexeis et al., 2018). The uncertainty of the BSFC for
different Euro class HDTs operating over a wide range of engine conditions is generally within 25% (Mahmoudzadeh Andwari
et al., 2017; He et al., 2017; Dreher and Harley, 1998; Heywood, 1988). Note that our measurements represent points of time
similar to those in a cycle where the particle emissions can be most prominent. Looking at a whole cycle, this value will be
averaged, hence the results from our instantaneous plume measurements may represent an upper limit of the emissions. In Fig.
2a, the right y-axis gives the EFs converted into units of g kWh$^{-1}$ and the Euro standards are shown as blue crosses.
In general, the measured median EF$_{PM}$ are lower than the Euro standards for all HDT types. In particular, the median
EF$_{PM}$ for Euro VI HDTs is more than one order of magnitude lower than the Euro standard regulation value since Diesel
Particulate Filters (DPFs) are required for these Euro VI HDTs to comply with PM and PN standards (Williams and Minjares,
2016). The effectiveness of DPF in reducing particle emissions have been confirmed by various studies (Martinet et al., 2017;
May et al., 2014; Mendoza-Villafuerte et al., 2017; Moody and Tate, 2017; Preble et al., 2015). For example, Bergmann et al.
(2009) illustrated that post-DPF PM concentrations decreased by 99.5% compared with pre-DPF for the New European Driving
Cycle (NEDC). In real-world measurements, at least ∼90% reductions in PM emissions compared to typical pre-DPF levels
have been reported (Bishop et al., 2015). Euro emission standards for PM of Euro IV and Euro V heavy-duty diesel engine are
the same, while the measured median EF$_{PM}$ of the Euro V fleet was around 1.5 times lower than that of the Euro IV fleet. The
control of diesel engine emissions typically requires a compromise between NO$_x$ and particle emission reduction (Clark et al.,
1999). The NO$_x$ emission standard is more stringent for Euro V compared to Euro IV (a factor of 43% lower), and hence Euro
V HDTs are generally equipped with SCR or EGR to reduce NO$_x$. In contrast, Euro IV engines are rarely equipped with NO$_x$
after-treatment systems and thus must achieve the NO$_x$ emission limits by tuning the engine performance parameters at the
expense of higher PM emissions (Preble et al., 2018; Van Setten et al., 2001). In each of the Euro III, Euro IV, Euro V, and
EEV classes, 25-50% of all the measured HDTs had an EF$_{PM}$ higher than their corresponding Euro standards. As described
previously, this comparison with the Euro standard is relative and indicative. The higher emissions from individual HDTs may
indicate deterioration of engine performance due to wear caused by aging, mileage accumulation, or inadequate maintenance.
Our study shows that Euro VI HDTs generally have low PM emissions, but HDTs from older Euro classes frequently exceeded
their PM emission limits, suggesting that improved maintenance and suitable retrofitting of older engines are needed.
For PN emissions, EF$_{PN}$ shows an overall trend similar to EF$_{PM}$. However, large data scatter was evident for Euro VI
HDTs, likely due to the sensitivity of nucleation mode particle formation to changes in driving conditions (Fig. 2b). Zheng et



al. (2014) reported high concentrations of nucleation mode particles under uphill driving conditions but low concentrations
under cruise and downhill driving conditions. It is important to note that the median $EF_{PN}$ of Euro VI HDTs was significantly
lower than those from the other Euro type HDTs, which indicates efficient PM removal by the DPF without compromising on
total PN emission. Nonetheless, the decrease of particle number in the accumulation mode removes particle surface area
available for condensation and therefore favours nucleation of organics from fuel and lubrication oil. Le Breton et al. (2019)
confirmed the contribution of lubrication oil in bus emissions. Besides, DPFs can act as a sulphur reservoir and when excess
sulphur is released, $SO_2$ to $SO_3$ conversion can take place once the after-treatment temperature reaches a critical level (Herner
et al., 2011). In this case, total particle number emissions can increase due to nucleation from gas-phase sulphuric acid. Since
the fuel sulphur content is low, more than 90% of Euro VI HDTs had an $EF_{SO2}$ lower than the threshold in this study, organics
would play a more important role in the formation of nucleation mode particles.

232        Figure 2c shows that the median $EF_{BC}$ was reduced by more than 99% for Euro VI HDTs compared to Euro III HDTs,

and the median $EF_{BC}$ of Euro VI HDTs was even at the threshold (0.2mg $(kg\ fuel)^{-1}$). The BC emissions generally showed an
overall decrease when moving towards newer Euro classes, which is similar to the $EF_{PM}$ trend with the exception of Euro IV
HDTs. Compared with Euro V HDTs, the median $EF_{BC}$ of Euro IV HDTs is 35% lower, however, the emission of the mixture
of BC and BrC from Euro IV HDTs is higher (Fig. S7a). Euro IV HDTs had the highest BrC contribution to the total light-
absorbing substances among all the Euro classes (Fig. S7b). Compared to the EEPS, the detection limit of the Aethalometer is
five times higher, which may influence the correlation between BC and PM at low mass loading conditions (Fig. S3).

239        Figure 2d compares the emissions of $NO_2$ and $NO_x$ from different Euro class HDTs. The vertical lines represent the

different Euro standards. HDTs with either $EF_{NO2}$ or $EF_{NOx}$ lower than the detection limits of the instruments were removed
from the figure. In general, Euro VI HDTs exhibit more than 90% reduction in both median and average $EF_{NOx}$ compared to
Euro III HDTs. This is consistent with Carslaw et al. (2011) who estimated a 93% $NO_x$ reduction from Euro III to Euro VI for
heavy goods vehicles (HGV) in the United Kingdom. Relatively, the Euro V HDTs had a larger fraction exceeding its Euro
standard, which may be due to the combined effects of poor engine tuning and the inactivity (low temperature) or deterioration
of SCR systems. Newer engines tend to exhibit a higher $NO_2$ emission fraction at a similar $NO_x$ level, and the Euro VI HDTs
show a relatively low median $EF_{NO2}$ with a large range of data scatter and several high emitters. A continuous increase of
$EF_{NO2}/EF_{NOx}$ was evident from Euro IV to Euro VI HDTs (Fig. S8b). This trend is consistent with Kozawa et al. (2014), who
reported an increase in the share of $NO_2$ to total $NO_x$ from Euro III to Euro V vehicles. Euro VI HDTs have a higher $NO_2$
fraction because the DOC upstream of the filter is used to convert NO to $NO_2$ to control the soot loading in the DPF and
facilitate the passive regeneration (Van Setten et al., 2001). A failure of the $NO_2$ reduction due to the inactivity of the SCR,
resulting from low exhaust gas temperature, may result in a higher $NO_2$ emission (Bishop et al., 2010; Heeb et al., 2010; Herner
et al., 2009; May et al., 2014; Thiruvengadam et al., 2015). A more significant decrease in $NO_x$ than $NO_2$ emissions of Euro
VI HDTs may cause an increase of $EF_{NO2}/EF_{NOx}$.

254        Compared with non-Swedish HDTs, Swedish HDTs generally have lower EFs in terms of all the pollutants (Fig. 2 and

Tables 1 and 2), which may be attributed to the more stringent domestic goals regarding pollution, clean air, greenhouse gas



emissions, energy efficiency, and innovative sustainable solutions (Government Offices of Sweden, 2017). One may note that
the non-Swedish HDTs was not identified according to Euro class and could contain a larger share of non-Euro VI trucks.
Table 1 compares the average emission data of PM and PN of the current work with previous studies according to the
HDT type and gives information on used measurement methods and driving conditions. Generally, the $EF_{PM}$ and $EF_{PN}$ in this
study are within the reported ranges of HDV emissions in the literature. Our estimated $EF_{PM}$ of Euro III HDTs are comparable
to those of Euro III buses in Hallquist et al. (2013) and Pirjola et al. (2016). HDTs and buses within the same Euro class emit
similar amounts of PM. Watne et al. (2018) show that DPF retrofitted Euro III buses have much lower particle EFs. While
$EF_{PM}$ is highly dependent on driving conditions such as speed and acceleration, the average $EF_{PM}$ of Euro IV, Euro V and EEV
HDTs of this study:172, 146, and 78 mg (kg fuel)$^{-1}$, respectively, are comparable to trends in previous studies (Hallquist et al.,
2013; Pirjola et al., 2016; Watne et al., 2018). Average $EF_{PM}$ of Euro VI HDTs (5 mg (kg fuel)$^{-1}$/ 1.1 mg km$^{-1}$) is within the
range of emissions from HDVs with DPFs, e.g., 0.6 - 20.5 mg km$^{-1}$ for a recent chassis dynamometer test (Jiang et al., 2018)
and 2.5 - 8.7 mg km$^{-1}$ for road measurements in California (Quiros et al., 2016). Note that size ranges and measurement
methodologies may differ among the studies as listed in Table 1. Since most of the particle emissions related to road traffic
combustion are below 560 nm (Fig. 4), the size range in our study is comparable to most other wider range PM measurements.
Larger particles from road measurements of total PM may include non-combustion-related particles, e.g. resuspended road
dust, tire and brake particles, and should be interpreted with caution. In contrast to $EF_{PM}$, a much less obvious decrease in
average $EF_{PN}$ was observed across different Euro classes. The reason for the high average particle emission for EEV and Euro
VI is likely due to high emissions of nucleation mode particles from a number of HDTs.
In Figure 3a, $EF_{PM}$ and $EF_{PN}$ of individual HDTs in this study and selected previous studies are plotted. HDTs with either
$EF_{PM}$ or $EF_{PN}$ lower than the detection limits of the instruments (0.07 mg (kg fuel)$^{-1}$ and $2.8 \times 10^{11}$# (kg fuel)$^{-1}$ respectively)
were removed from the figure (24% of the data). Generally, both $EF_{PM}$ and $EF_{PN}$ exhibited a decreasing trend from Euro III to
Euro IV and from Euro V to EEV HDTs. Overall, Euro VI HDTs had drastically lower PM emissions but highly scattered PN
emissions. Older Euro type buses retrofitted with DPF were shown to have reduced particle emissions, and some retrofitted
Euro III buses (black open triangles in Fig. 3a) may perform as well as Euro VI HDTs, indicating the effectiveness of
retrofitting older HDTs.
In more recent Euro standards, PN regulation has been introduced. The SPN as defined by the European Particle
Measurement Program (PMP) is the number of particles which remain after passing through an evaporation tube with a wall
temperature of 300-400°C (Zheng et al., 2011). The PMP only measures and regulates solid particles with a diameter larger
than 23 nm because measurements of smaller particles in the nucleation mode have poor repeatability (Martini et al., 2009).
SPN larger than 23 nm was integrated into the European emission regulation in 2013 for Euro VI heavy-duty
engines (Giechaskiel et al., 2012). A potential issue of evaporation measurements is that a fraction of the sub-23 nm particles
can also be formed downstream of the European PMP methodology through re-nucleation of semi-volatile precursors (Zheng
et al., 2012; Zheng et al., 2011). In our study, the TD temperature of 250°C is lower than the maximum temperature used by
the PMP (300-400°C) and does not follow the exact operation specifications of the PMP. However, Amanatidis et al. (2018)



summarised that TD is a suitable alternative approach for the removal of volatile particles. Particles larger than 23 nm downstream of the TD were measured by the EEPS and we integrated the size bins from 23.5 nm to 560 nm to represent the SPN. Figure 3b compares the $EF_{PM}$ and $EF_{SPN}$ of Euro VI HDTs. Generally, after-treatment control systems could not reduce SPN emissions as effectively as PM emissions, indicating that more control of SPN emission of Euro VI HDTs may be necessary.

Shown in Table 2 are the average EFs of gaseous pollutants ($NO_x$, CO, HC) in this study compared with other studies. $EF_{NOx}$ generally decreased from the Euro III (43.3 g (kg fuel)$^{-1}$) to Euro VI (3.1 g (kg fuel)$^{-1}$) class, and are in good agreement with reported values for HDTs in the literature. $EF_{NOx}$ of Euro III HDTs is moderately higher in this study. Note that the $EF_{NOx}$ and $EF_{PM}$ of EEV were much higher in Pirjola et al. (2016), in which only a limited number (3-4) vehicles were tested and hard braking was common in approaching a 90° turn before accelerating again. The ratio of $EF_{NO2}$ to $EF_{NOx}$ generally agrees with the projection in Kousoulidou et al. (2008) and on-road plume chasing measurements in Lau et al. (2015), while the ratio is lower for the older Euro class HDTs compared with the remote sensing study in the UK (Carslaw and Rhys-Tyler, 2013). Carslaw et al. (2019) reported a decreasing trend of $EF_{NO2}/EF_{NOx}$ with vehicle mileage for Euro 6 light-duty diesel vehicles, while no significant trend was identified for Euro VI HDTs in this study. There may be other parameters influencing the $NO_x$ emission. For example, Ko et al. (2019) reported that the $NO_x$ emissions from Euro VI diesel vehicles were 29% higher in a traffic jam than in smooth traffic conditions. The temperature of the exhaust and DPF regeneration may also influence the $EF_{NOx}$.

Compared with $EF_{NOx}$, $EF_{CO}$ decreased less pronounced from Euro III to Euro VI HDTs (57%). Compared with newer Euro class HDTs, a larger fraction of HDTs in older Euro classes have an $EF_{CO}$ exceeding the Euro standards, which indicates that engine deterioration may have a serious effect on the CO emissions (Fig. S8c). Hallquist et al. (2013) reported a positive relationship between $EF_{CO}$ and $EF_{PM}$, i.e. high CO indicates incomplete combustion which favors soot formation. DPFs may also reduce CO in addition to PM (Hallquist et al., 2013), which is in agreement with the lowest CO emission of 15.5 g (kg fuel)$^{-1}$observed for DPF equipped Euro VI HDTs in this study. HC emission was relatively low for all HDT types, but no obvious decreasing trend was evident for $EF_{HC}$ from Euro III to Euro VI HDTs (Fig. S8d and Table 2). This does not reflect the more stringent Euro standard limit regarding HC where the Euro VI limit is more than a factor of three lower than the preceding Euro V/IV standards.

The 46 Swedish HDTs without available Euro type information emitted similar levels of particle and gaseous pollutants to Euro VI HDTs and were thus likely equipped with newer Euro engines.

**3.4 Size-resolved $EF_{PN}$ of volatile and non-volatile particles**

Figure 4 shows the average size-resolved number emission factors (solid lines) simultaneously measured via the bypass and TD lines for different Euro class HDTs. The $EF_{PN}$ of the volatile components is calculated as the difference of $EF_{PN}$ measured after the bypass line and the non-volatile component $EF_{PN}$ measured after the TD line. To differentiate between nucleation and accumulation mode particles, a cut point particle diameter of 30 nm was used as defined by Kittelson et al. (2002). In general,





all Euro III, Euro IV, Euro V and EEV HDTs showed a bimodal particle number size distribution, with one mode peaking at
~6-10 nm (nucleation mode) and another at ~50-80 nm (accumulation/soot mode) (Maricq, 2007). For Euro VI HDTs the
particle number size distributions were dominated by the nucleation mode. The $EF_{PN}$ of the accumulation mode particles
showed a decreasing trend from Euro III to EEV and for the Euro VI HDTs the accumulation mode was insignificant. For
heavy-duty diesel engines without a particulate filter, nucleation mode particles are mainly formed from organics. For vehicles
with DPF both organics and the fuel sulphur content influence the formation of nucleation mode particles (Vaaraslahti et al.,
2004). Thiruvengadam et al. (2012) found a direct relationship between exhaust nanoparticles in the nucleation mode and the
exhaust temperature of the DPF-SCR equipped diesel engine. These factors lead to high variability in the nucleation mode
fraction of $EF_{PN}$. Most particles in the nucleation mode evaporate after passing through the TD. Sakurai et al. (2003b) reported
that volatile compounds in diesel particles are mainly comprised of unburned lubrication oil.
The non-volatile components in the nucleation mode may consist of metallic ash from lubrication oil or fuel
additives (Sakurai et al., 2003a) or some organic compounds of extremely low volatility (Gkatzelis et al., 2016). In the
accumulation mode, the particle mode diameter shifted towards lower sizes after passing the TD. In Fig. 4, we also present the
median size distribution (dashed lines). There is a small difference between mean and median size distributions in the
accumulation mode while a bigger difference occurs in the nucleation mode. The latter mode is more dynamic and there are
larger possibilities for extreme values skewing the averages.
To be consistent with previous studies which overwhelmingly report average size distributions, we choose to utilize the
average size distributions for the discussions below. The volatilities of particle emissions in the accumulation and nucleation
mode have been evaluated by calculating the average $EF_{PN}$ and $EF_{PM}$ fraction remaining (after heating) of particles emitted
from Euro III-VI HDTs (Fig. S9). In general, the $EF_{PN}$ fraction remaining in the nucleation mode was lower than that in the
accumulation mode across all HDTs in all Euro classes. In terms of particle mass, the nucleation mode and accumulation mode
showed similar $EF_{PM}$ fractions remaining from Euro III to EEV HDTs, while Euro VI HDTs had a much lower $EF_{PM}$ fraction
remaining in the nucleation mode than in the accumulation mode. Compared with other Euro class HDTs, Euro VI HDTs had
the lowest $EF_{PN}$ and $EF_{PM}$ fraction remaining in both nucleation and accumulation mode. Around 94% of the particles by
number and 55% of the particles by mass (or volume) in total were evaporated. Alfoldy et al. (2009) reported that if the same
amount of volatile mass in the nucleation mode and accumulation mode were inhaled, 48% and 29% of the mass would deposit
in the lung respectively, implying that volatile mass in the nucleation mode would exert a 1.5 times stronger effect.

**3.5 Emissions from high emitters**

Figure 5 shows the cumulative emission distributions for PM, PN, $NO_x$ and $NO_2$ emissions, with HDTs ranked in order from
dirtiest to cleanest. The plots show a significant skewedness towards a small fraction of HDTs with a high fraction of total
emissions (deviation from 1:1 line) for each pollutant, indicating the importance of "high emitters". The disproportionate
skewed distribution of pollutants is a common feature of on-road emission measurements (Preble et al., 2018; Preble et al.,
2015; Dallmann et al., 2012). The highest-emitting 10% of HDTs in each pollutant were responsible for 65% of total PM, 70%



of total PN, 44% of total $NO_x$ emissions and 69% of total $NO_2$, respectively. The distribution of $NO_x$ has the least skewedness
compared with the other pollutants. If the 10% highest emitters for each pollutant were removed, the corresponding average
EF for PM, PN, $NO_x$, and $NO_2$ would decrease by 62%, 67%, 38%, and 66% respectively. However, the high emitters for each
pollutant are different. For example, Euro III HDTs account for 70% and 67% of the top 3% emitters for PM and BC emissions,
while Euro VI HDTs account for 80% and 56% of the top 3% emitters for PN and $NO_2$ emissions. Here, top 3% emitters were
chosen as the reference because Euro III HDTs only accounted for 3% of the total number of HDTs. Lau et al. (Lau et al.,
2015) similarly reported that not all high-emitters were members of the lower Euro classes and that high-emitters for a
particular pollutant may not simultaneously be high-emitters for other pollutants.

**3.6 Fleet characteristics**

Figure 6a-d show the changes in average EFs of PM, PN, BC, and $NO_x$ with respect to the registration year of the HDTs.
Triennial average EFs were calculated, with truck registration years divided into 5 bins (2002-2005, 2006-2008, 2009-2011,
2012-2014 and 2015-2017). The black arrows in Fig. 6d show the years that the particular type of HDTs examined in this study
was first registered. Coupled with the possible phase-out of older fleets, the HDTs with more advanced engines gradually
accounted for a higher proportion of the total fleet. There is a significant improvement during the last years and the transition
to widespread adoption of Euro VI will take real-world on-road emissions into a new era of much lower contributions to air
pollution.

372        To estimate for each Euro class the typical contribution of air pollution emissions we utilised the number of kilometres

driven by HDTs on Swedish roads. During 2018, $4.1 \times 10^9$ and $9.2 \times 10^8$ kilometres were driven by Swedish and non-Swedish
diesel HDTs on Swedish roads, contributing to 82% and 18% to the total distances travelled by diesel HDTs respectively (Fig.
7a). The numbers of kilometres driven by Swedish Euro 0, Euro I, Euro II, Euro III, Euro IV, Euro V and Euro VI diesel HDTs
were $2.8 \times 10^7$, $5.0 \times 10^6$, $5.4 \times 10^7$, $2.0 \times 10^8$, $3.1 \times 10^8$, $1.3 \times 10^9$ and $2.2 \times 10^9$ respectively (HBEFA 3.3, 2019). In Figure 7b, the
relative contributions of kilometers driven by Swedish Euro 0 to Euro VI HDTs are shown. Zhang et al. (2014) reported no
statistically significant difference in fuel consumption among Euro II to Euro IV buses under a real-world typical bus driving
cycle in Beijing. In this study, we assume the fuel consumption per kilometre and fuel density are the same across the different
Euro class HDTs and adopting the average fuel-based EFs calculated in this study (Tables 1 and 2), the approximation of
contributions of pollutants emitted from Swedish HDTs in each Euro class to the total PM, PN, BC and $NO_x$ emissions are
depicted in Fig. 7c-f. Due to a lack of corresponding emission information, pollutant average EFs of Euro 0, Euro I and Euro
II HDTs were assumed to be at the same level as those of Euro III HDTs representing lower bound estimates. Euro 0-II HDTs
accounted for less than 2.2% of the grand total distance driven but totally contributed to 16%, 13%, 6% and 4% of BC, PM,
$NO_x$, and PN emissions. Euro III HDTs only accounted for 5% (Fig. 7b) of the total fleet but disproportionally contributed to
37%, 30%, 16% and 10% of BC, PM, $NO_x$, and PN emissions. Euro IV HDTs also exhibited disproportionally high PM and
$NO_x$ emissions. A fraction of 32% of HDTs belonged to the Euro V category, they contributed to 53%, 42%, 34%, and 32%
of $NO_x$, PM, BC and PN emissions respectively. Upgrading, replacing or making obsolete Euro 0 to Euro V HDTs would be





necessary for mitigating a large part of the PM, PN, BC and NO$_x$ emissions. Euro VI HDTs accounted for the highest fraction
of the total fleet (53%), but only contributed to 2%, 6%, 12% and 47% of PM, BC, NO$_x$, and PN emissions, indicating
successful overall pollution reduction with the introduction of more Euro VI HDVs. Using the median EFs as references, the
emission contributions from Euro VI HDTs to the total pollutant emissions would be even lower (Fig. S10). These data provide
useful information to predict future pollutant emission trends and to guide policy analysis and implementation. Since the
predicted transition in emissions from road transport would be significant, chemical transport model/cost-assessment models
need to get fast access to emission factors for new generation HDTs to be able to provide a better estimation of near future air
pollution levels.
**4 Atmospheric implications and conclusions**
The transition in the atmospheric emission of particles and gases from on-road HDTs caused by the modernisation of the fleet
is reported in this study. Particle emissions of PM, BC and to a lesser extent PN exhibited substantial reductions from Euro III
to Euro VI HDTs. The gaseous emissions of NO$_x$ and CO also showed significant decrease with respect to Euro class, while
the HC emission was relatively low for all the HDT Euro class types. Compared with Euro III HDTs, Euro VI HDTs showed
99%, 98%, 93% and 57% reductions of the average emissions factors of PM, BC, NO$_x$, and CO respectively. Although a
significant reduction in NO$_x$ emissions and a lower median EF$_{NO2}$ were evident, the fraction of NO$_2$ in the NO$_x$ emissions
increased continuously from Euro IV to Euro VI HDTs, and Euro VI HDTs were the dominant class of the top 3% emitters
for NO$_2$. PN showed the largest data scattering for Euro VI HDTs, though after evaporation of the volatile fraction, SPN
became less scattered. A plausible reason for this large variability in PN but not PM is the formation of nucleation mode
particles containing more volatile compounds, which is more sensitive to individual driving and plume dilution conditions.
Driving condition and engine technology affected the size distribution of particle number emissions. The average particle
number size distributions of Euro III to EEV HDTs were bimodal with nucleation modes at ~6-10 nm and accumulation modes
at ~50-80 nm. Euro VI HDTs displayed nucleation mode dominant size distributions. Measurements of particle volatility
revealed that Euro VI HDTs had the highest volatile fraction in both nucleation mode and accumulation mode compared to the
other Euro classes. More detailed chemical composition information of this volatile fraction is needed to assess their potential
impacts for health and formation of SOA.
We also found that a small number of high emitters contributed to a large fraction of the total emissions. The top 10%
emitters in each pollutant category were responsible for 65% of total PM, 70% of total PN, 44% of total NO$_x$ and 69% of total
NO$_2$ emissions, respectively. Euro III HDTs were the dominant top 3% emitters for PM and BC emissions, and Euro VI HDTs
were the dominant top 3% emitters for PN and NO$_2$ emissions.
In general, an overall pollution reduction has been achieved during the last years and the transition to Euro VI adoption
will take real-world on-road emissions into a new era of much lower contributions to air pollution. The emissions of PM, BC
and NO$_x$ are predicted to further decrease in the future, while PN emissions may be subject to greater fluctuation and therefore





421 be more challenging to control. Upgrading or phasing-out of existing Euro 0 to Euro V vehicles and introducing more Euro VI

422 HDTs would result in large pollution reductions. More intensive attentions need to be focused on SPN controls for Euro VI

423 HDTs. A careful and more detailed examination of the impacts of fleet upgrades in terms of ambient pollutant levels and

424 emission reduction targets for individual pollutants may be needed for further evaluation.

425

426 *Data availability.*

427 The data used in this publication are available to the community and can be accessed by request to the corresponding author.

428 *Author contributions.*

429 ÅMH designed the project; LZ, ÅMH, CMS, SMG, ÅS, MJ, HS, MH, and IW conducted the measurements; LZ, CMS, and

430 QL analysed data; LZ, ÅMH, MH, CKC and BPL wrote the paper. All co-authors contributed to the discussion of the

431 manuscript.

432 *Competing interests.*

433 The authors declare that they have no conflict of interest.

434 *Acknowledgments.*

435 This work was financed by Formas (214-2013-1430) and was an initiative within the framework programme "Photochemical

436 smog in China" financed by the Swedish Research Council (639-2013-6917). Chak K. Chan would like to acknowledge the

437 support of the Science Technology and Innovation Committee of Shenzhen Municipality (project no.

438 JCYJ20160401095857424).

439

440 **References**

441 Alfoldy, B., Giechaskiel, B., Hofmann, W., and Drossinos, Y.: Size-distribution dependent lung deposition of diesel exhaust
442 particles, J. Aerosol Sci., 40, 652-663, 10.1016/j.jaerosci.2009.04.009, 2009.

443 Amanatidis, S., Ntziachristos, L., Karjalainen, P., Saukko, E., Simonen, P., Kuittinen, N., Aakko-Saksa, P., Timonen, H.,
444 Ronkko, T., and Keskinen, J.: Comparative performance of a thermal denuder and a catalytic stripper in sampling laboratory
445 and marine exhaust aerosols, Aerosol Sci. Tech., 52, 420-432, https://doi.org/10.1080/02786826.2017.1422236, 2018.

446 Arnold, F., Pirjola, L., Ronkko, T., Reichl, U., Schlager, H., Lahde, T., Heikkila, J., and Keskinen, J.: First online
447 measurements of sulfuric acid gas in modern heavy-duty diesel engine exhaust: implications for nanoparticle formation,
448 Environ. Sci. Technol., 46, 11227-11234, https://doi.org/10.1021/es302432s, 2012.

449 Ban-Weiss, G. A., Lunden, M. M., Kirchstetter, T. W., and Harley, R. A.: Measurement of black carbon and particle number
450 emission factors from individual heavy-duty trucks, Environ. Sci. Technol., 43, 1419-1424, https://doi.org/10.1021/es8021039,
451 2009.





Bergmann, M., Kirchner, U., Vogt, R., and Benter, T.: On-road and laboratory investigation of low-level PM emissions of a
modern diesel particulate filter equipped diesel passenger car, Atmos. Environ., 43, 1908-1916,
https://doi.org/10.1016/j.atmosenv.2008.12.039, 2009.
Bishop, G. A., and Stedman, D. H.: A decade of on-road emissions measurements, Environ. Sci. Technol., 42, 1651-1656,
https://doi.org/10.1021/es702413b, 2008.
Bishop, G. A., Peddle, A. M., Stedman, D. H., and Zhan, T.: On-road emission measurements of reactive nitrogen compounds
from three California cities, Environ. Sci. Technol., 44, 3616-3620, https://doi.org/10.1021/es903722p, 2010.
Bishop, G. A., Schuchmann, B. G., and Stedman, D. H.: Heavy-duty truck emissions in the South Coast Air Basin of California,
Environ. Sci. Technol., 47, 9523-9529, https://doi.org/10.1021/es401487b, 2013.
Bishop, G. A., Hottor-Raguindin, R., Stedman, D. H., McClintock, P., Theobald, E., Johnson, J. D., Lee, D. W., Zietsman, J.,
and Misra, C.: On-road heavy-duty vehicle emissions monitoring system, Environ. Sci. Technol., 49, 1639-1645,
https://doi.org/10.1021/es505534e, 2015.
Biswas, S., Verma, V., Schauer, J. J., Cassee, F. R., Cho, A. K., and Sioutas, C.: Oxidative potential of semi-volatile and non-
volatile particulate matter (PM) from heavy-duty vehicles retrofitted with emission control technologies, Environ. Sci.
Technol., 43, 3905-3912, https://doi.org/10.1021/es9000592, 2009.
Bocchi, C., Bazzini, C., Fontana, F., Pinto, G., Martino, A., and Cassoni, F.: Characterization of urban aerosol: seasonal
variation of mutagenicity and genotoxicity of PM2.5, PM1 and semi-volatile organic compounds, Mutat. Res., 809, 16-23,
https://doi.org/10.1016/j.mrgentox.2016.07.007, 2016.
Burgard, D. A., Bishop, G. A., Stedman, D. H., Gessner, V. H., and Daeschlein, C.: Remote sensing of in-use heavy-duty
diesel trucks, Environ. Sci. Technol., 40, 6938-6942, https://doi.org/10.1021/es060989a, 2006.
Burgard, D. A., and Provinsal, M. N.: On-road, in-use gaseous emission measurements by remote sensing of school buses
equipped with diesel oxidation catalysts and diesel particulate filters, J. Air Waste Manag. Assoc., 59, 1468-1473,
https://doi.org/10.3155/1047-3289.59.12.1468, 2009.
Campagnolo, D., Cattaneo, A., Corbella, L., Borghi, F., Del Buono, L., Rovelli, S., Spinazze, A., and Cavallo, D. M.: In-
vehicle airborne fine and ultra-fine particulate matter exposure: The impact of leading vehicle emissions, Environ. Int., 123,
407-416, https://doi.org/10.1016/j.envint.2018.12.020, 2019.
Carslaw, D. C., Beevers, S. D., Tate, J. E., Westmoreland, E. J., and Williams, M. L.: Recent evidence concerning higher NOx
emissions from passenger cars and light-duty vehicles, Atmos. Environ., 45, 7053-7063,
https://doi.org/10.1016/j.atmosenv.2011.09.063, 2011.
Carslaw, D. C., and Rhys-Tyler, G.: New insights from comprehensive on-road measurements of NOx, NO2 and NH3 from
vehicle emission remote sensing in London, UK, Atmos. Environ., 81, 339-347,
https://doi.org/10.1016/j.atmosenv.2013.09.026, 2013.
Carslaw, D. C., Farren, N. J., Vaughan, A. R., Drysdale, W. S., Young, S., and Lee, J. D.: The diminishing importance of
nitrogen dioxide emissions from road vehicle exhaust, Atmospheric Environment: X, 1, 100002,
https://doi.org/10.1016/j.aeaoa.2018.100002, 2019.
Chen, L., Wang, Z., Liu, S., and Qu, L.: Using a chassis dynamometer to determine the influencing factors for the emissions
of Euro VI vehicles, Transport. Res. D- Tr. E., 65, 564-573, https://doi.org/10.1016/j.trd.2018.09.022, 2018.



Cheung, H. H. Y., Tan, H. B., Xu, H. B., Li, F., Wu, C., Yu, J. Z., and Chan, C. K.: Measurements of non-volatile aerosols
with a VTDMA and their correlations with carbonaceous aerosols in Guangzhou, China, Atmos.
Chem. Phys., 16, 8431-8446, https://doi.org/10.5194/acp-16-8431-2016, 2016.
Clark, N. N., Gautam, M., Rapp, B. L., Lyons, D. W., Graboski, M. S., McCormick, R. L., Alleman, T. L., and Norton, P.:
Diesel and CNG transit bus emissions characterization by two chassis dynamometer laboratories: Results and issues, SAE
Technical Paper, 0148-7191, 1999.
Cui, M., Chen, Y. J., Feng, Y. L., Li, C., Zheng, J. Y., Tian, C. G., Yan, C. Q., and Zheng, M.: Measurement of PM and its
chemical composition in real-world emissions from non-road and on-road diesel vehicles, Atmos.
Chem. Phys., 17, 6779-6795, https://doi.org/10.5194/acp-17-6779-2017, 2017.
Dallmann, T. R., DeMartini, S. J., Kirchstetter, T. W., Herndon, S. C., Onasch, T. B., Wood, E. C., and Harley, R. A.: On-road
measurement of gas and particle-phase pollutant emission factors for individual heavy-duty diesel trucks, Environ. Sci.
Technol., 46, 8511-8518, https://doi.org/10.1021/es301936c, 2012.
Donahue, N. M., Robinson, A. L., Stanier, C. O., and Pandis, S. N.: Coupled partitioning, dilution, and chemical aging of
semivolatile organics, Environ. Sci. Technol., 40, 2635-2643, https://doi.org/10.1021/es052297c, 2006.
Dreher, D. B., and Harley, R. A.: A fuel-based inventory for heavy-duty diesel truck emissions, J. Air Waste Manage., 48,
352-358, https://doi.org/10.1080/10473289.1998.10463686, 1998.
Du, Y., Xu, X., Chu, M., Guo, Y., and Wang, J.: Air particulate matter and cardiovascular disease: the epidemiological,
biomedical and clinical evidence, J. Thorac. Dis., 8, E8-E19, http://dx.doi.org/10.3978/j.issn.2072-1439.2015.11.37 2016.
Edwards, R., Larivé, J., Rickeard, D., and Weindorf, W.: Well-to-Wheels analysis of future automotive fuels and powertrains
in the European context: Well-to-Tank Appendix 2-Version 4a, Joint Research Centre of the European Commission, EUCAR,
and CONCAWE, 1-133, http://10.2790/95629, 2014.
EEA: Air Quality in Europe - 2018 Report, 2018.
Franco, V., Kousoulidou, M., Muntean, M., Ntziachristos, L., Hausberger, S., and Dilara, P.: Road vehicle emission factors
development: A review, Atmos. Environ., 70, 84-97, https://doi.org/10.1016/j.atmosenv.2013.01.006, 2013.
Gentner, D. R., Jathar, S. H., Gordon, T. D., Bahreini, R., Day, D. A., El Haddad, I., Hayes, P. L., Pieber, S. M., Platt, S. M.,
de Gouw, J., Goldstein, A. H., Harley, R. A., Jimenez, J. L., Prevot, A. S., and Robinson, A. L.: Review of Urban Secondary
Organic Aerosol Formation from Gasoline and Diesel Motor Vehicle Emissions, Environ. Sci. Technol., 51, 1074-1093,
https://doi.org/10.1021/acs.est.6b04509, 2017.
Gertler, A. W.: Diesel vs. gasoline emissions: Does PM from diesel or gasoline vehicles dominate in the US?, Atmos. Environ.,
39, 2349-2355, https://doi.org/10.1016/j.atmosenv.2004.05.065, 2005.
Giechaskiel, B., Alfoldy, B., and Drossinos, Y.: A metric for health effects studies of diesel exhaust particles, J. Aerosol Sci.,
40, 639-651, https://doi.org/10.1016/j.jaerosci.2009.04.008, 2009.
Giechaskiel, B., Mamakos, A., Andersson, J., Dilara, P., Martini, G., Schindler, W., and Bergmann, A.: Measurement of
Automotive Nonvolatile Particle Number Emissions within the European Legislative Framework: A Review, Aerosol Sci.
Tech., 46, 719-749, https://doi.org/10.1080/02786826.2012.661103, 2012.



Giechaskiel, B., Schwelberger, M., Delacroix, C., Marchetti, M., Feijen, M., Prieger, K., Andersson, S., and Karlsson, H. L.: Experimental assessment of solid particle number Portable Emissions Measurement Systems (PEMS) for heavy-duty vehicles applications, J. Aerosol Sci., 123, 161-170, https://doi.org/10.1016/j.jaerosci.2018.06.014, 2018.

Gkatzelis, G. I., Papanastasiou, D. K., Florou, K., Kaltsonoudis, C., Louvaris, E., and Pandis, S. N.: Measurement of nonvolatile particle number size distribution, Atmos. Meas. Tech., 9, 103-114, https://doi.org/10.5194/amt-9-103-2016, 2016.

The climate policy framework: http://www.government.se/articles/2017/06/the-climate-policy-framework/, 2017.

Grigoratos, T., Fontaras, G., Giechaskiel, B., and Zacharof, N.: Real-world emissions performance of heavy-duty Euro VI diesel vehicles, Atmos. Environ., 201, 348-359, https://doi.org/10.1016/j.atmosenv.2018.12.042, 2019.

Guo, J. D., Ge, Y. S., Hao, L. J., Tan, J. W., Li, J. Q., and Feng, X. Y.: On-road measurement of regulated pollutants from diesel and CNG buses with urea selective catalytic reduction systems, Atmos. Environ., 99, 1-9, https://doi.org/10.1016/j.atmosenv.2014.07.032, 2014.

Hak, C. S., Hallquist, M., Ljungstrom, E., Svane, M., and Pettersson, J. B. C.: A new approach to in-situ determination of roadside particle emission factors of individual vehicles under conventional driving conditions, Atmos. Environ., 43, 2481-2488, https://doi.org/10.1016/j.atmosenv.2009.01.041, 2009.

Hallquist, A. M., Jerksjo, M., Fallgren, H., Westerlund, J., and Sjodin, A.: Particle and gaseous emissions from individual diesel and CNG buses, Atmos. Chem. Phys., 13, 5337-5350, https://doi.org/10.5194/acp-13-5337-2013, 2013.

Hallquist, M., Wenger, J. C., Baltensperger, U., Rudich, Y., Simpson, D., Claeys, M., Dommen, J., Donahue, N. M., George, C., Goldstein, A. H., Hamilton, J. F., Herrmann, H., Hoffmann, T., Iinuma, Y., Jang, M., Jenkin, M. E., Jimenez, J. L., Kiendler-Scharr, A., Maenhaut, W., McFiggans, G., Mentel, T. F., Monod, A., Prevot, A. S. H., Seinfeld, J. H., Surratt, J. D., Szmigielski, R., and Wildt, J.: The formation, properties and impact of secondary organic aerosol: current and emerging issues, Atmos. Chem. Phys., 9, 5155-5236, https://doi.org/10.5194/acp-9-5155-2009, 2009.

Haugen, M. J., Bishop, G. A., Thiruvengadam, A., and Carder, D. K.: Evaluation of Heavy-and Medium-Duty On-Road Vehicle Emissions in California's South Coast Air Basin, Environ. Sci. Technol., 52, 13298-13305, https://doi.org/10.1021/acs.est.8b03994, 2018.

He, C., Li, J., Ma, Z., Tan, J., and Zhao, L.: High NO2/NOx emissions downstream of the catalytic diesel particulate filter: An influencing factor study, J. Environ. Sci. (China), 35, 55-61, https://doi.org/10.1016/j.jes.2015.02.009, 2015.

He, L., Hu, J., Zhang, S., Wu, Y., Guo, X., Guo, X., Song, J., Zu, L., Zheng, X., and Bao, X.: Investigating real-world emissions of China's heavy-duty diesel trucks: Can SCR effectively mitigate NOx emissions for highway trucks, Aerosol Air. Qual. Res., 17, 2585-2594, https://DOI: 10.4209/aaqr.2016.12.0531, 2017.

Heeb, N. V., Schmid, P., Kohler, M., Gujer, E., Zennegg, M., Wenger, D., Wichser, A., Ulrich, A., Gfeller, U., Honegger, P., Zeyer, K., Emmenegger, L., Petermann, J. L., Czerwinski, J., Mosimann, T., Kasper, M., and Mayer, A.: Impact of low- and high-oxidation diesel particulate filters on genotoxic exhaust constituents, Environ. Sci. Technol., 44, 1078-1084, https://doi.org/10.1021/es9019222, 2010.

Herner, J. D., Hu, S., Robertson, W. H., Huai, T., Collins, J. F., Dwyer, H., and Ayala, A.: Effect of advanced aftertreatment for PM and NO x control on heavy-duty diesel truck emissions, Environ. Sci. Technol., 43, 5928-5933, https://doi.org/10.1021/es9008294, 2009.





Herner, J. D., Hu, S., Robertson, W. H., Huai, T., Chang, M.-C. O., Rieger, P., and Ayala, A.: Effect of advanced aftertreatment
for PM and NO x reduction on heavy-duty diesel engine ultrafine particle emissions, Environ. Sci. Technol., 45, 2413-2419,
https://doi.org/10.1021/es102792y, 2011.
Heywood, J. B.: Internal combustion engine fundamentals, 1988.
Huffman, J. A., Ziemann, P. J., Jayne, J. T., Worsnop, D. R., and Jimenez, J. L.: Development and characterization of a fast-
stepping/scanning thermodenuder for chemically-resolved aerosol volatility measurements, Aerosol Sci. Tech., 42, 395-407,
https://doi.org/10.1080/02786820802104981, 2008.
Janhall, S., and Hallquist, M.: A novel method for determination of size-resolved, submicrometer particle traffic emission
factors, Environ. Sci. Technol., 39, 7609-7615, https://doi.org/10.1021/es048208y, 2005.
Jarvinen, A., Timonen, H., Karjalainen, P., Bloss, M., Simonen, P., Saarikoski, S., Kuuluvainen, H., Kalliokoski, J., Dal Maso,
M., Niemi, J. V., Keskinen, J., and Ronklo, T.: Particle emissions of Euro VI, EEV and retrofitted EEV city buses in real
traffic, Environ. Pollut., 250, 708-716, https://doi.org/10.1016/j.envpol.2019.04.033, 2019.
Jiang, Y., Yang, J., Cocker, D., 3rd, Karavalakis, G., Johnson, K. C., and Durbin, T. D.: Characterizing emission rates of
regulated pollutants from model year 2012+ heavy-duty diesel vehicles equipped with DPF and SCR systems, Sci. Total
Environ., 619-620, 765-771, https://doi.org/10.1016/j.scitotenv.2017.11.120, 2018.
Kittelson, D., Watts, W., and Johnson, J.: Diesel Aerosol Sampling Methodology–CRC E-43, Final report, Coordinating
Research Council, 2002.
Ko, J., Myung, C. L., and Park, S.: Impacts of ambient temperature, DPF regeneration, and traffic congestion on NOx emissions
from a Euro 6-compliant diesel vehicle equipped with an LNT under real-world driving conditions, Atmos. Environ., 200, 1-
14, https://doi.org/10.1016/j.atmosenv.2018.11.029, 2019.
Kousoulidou, M., Ntziachristos, L., Mellios, G., and Samaras, Z.: Road-transport emission projections to 2020 in European
urban environments, Atmos. Environ., 42, 7465-7475, https://doi.org/10.1016/j.atmosenv.2008.06.002, 2008.
Kozawa, K. H., Park, S. S., Mara, S. L., and Herner, J. D.: Verifying emission reductions from heavy-duty diesel trucks
operating on Southern California freeways, Environ. Sci. Technol., 48, 1475-1483, https://doi.org/10.1021/es4044177, 2014.
Lau, C. F., Rakowska, A., Townsend, T., Brimblecombe, P., Chan, T. L., Yam, Y. S., Mocnik, G., and Ning, Z.: Evaluation
of diesel fleet emissions and control policies from plume chasing measurements of on-road vehicles, Atmos. Environ., 122,
171-182, https://doi.org/10.1016/j.atmosenv.2015.09.048, 2015.
Le Breton, M., Psichoudaki, M., Hallquist, M., Watne, Å., Lutz, A., and Hallquist, Å.: Application of a FIGAERO ToF CIMS
for on-line characterization of real-world fresh and aged particle emissions from buses, Aerosol Sci. Tech., 1-16,
https://doi.org/10.1080/02786826.2019.1566592, 2019.
Lee, B. H., Lopez-Hilfiker, F. D., Mohr, C., Kurten, T., Worsnop, D. R., and Thornton, J. A.: An iodide-adduct high-resolution
time-of-flight chemical-ionization mass spectrometer: application to atmospheric inorganic and organic compounds, Environ.
Sci. Technol., 48, 6309-6317, https://doi.org/10.1021/es500362a, 2014.
Li, X., Dallmann, T. R., May, A. A., Stanier, C. O., Grieshop, A. P., Lipsky, E. M., Robinson, A. L., and Presto, A. A.: Size
distribution of vehicle emitted primary particles measured in a traffic tunnel, Atmos. Environ., 191, 9-18,
https://doi.org/10.1016/j.atmosenv.2018.07.052, 2018.





Liu, Q., Hallquist, Å. M., Fallgren, H., Jerksjö, M., Jutterström, S., Salberg, H., Hallquist, M., Le Breton, M., Pei, X., and Pathak, R. K.: Roadside assessment of a modern city bus fleet: Gaseous and particle emissions, Atmospheric Environment: X, 100044, https://doi.org/10.1016/j.aeaoa.2019.100044, 2019.

Liu, Z. G., Vasys, V. N., Dettmann, M. E., Schauer, J. J., Kittelson, D. B., and Swanson, J.: Comparison of Strategies for the Measurement of Mass Emissions from Diesel Engines Emitting Ultra-Low Levels of Particulate Matter, Aerosol Sci. Tech., 43, 1142-1152, https://doi.org/10.1080/02786820903219035, 2009.

Lv, Y., Li, X., Xu, T. T., Cheng, T. T., Yang, X., Chen, J. M., Iinuma, Y., and Herrmann, H.: Size distributions of polycyclic aromatic hydrocarbons in urban atmosphere: sorption mechanism and source contributions to respiratory deposition, Atmos. Chem. Phys., 16, 2971-2983, https://doi.org/10.5194/acp-16-2971-2016, 2016.

Mahmoudzadeh Andwari, A., Pesiridis, A., Esfahanian, V., Salavati-Zadeh, A., Karvountzis-Kontakiotis, A., and Muralidharan, V.: A comparative study of the effect of turbo compounding and ORC waste heat recovery systems on the performance of a turbocharged heavy-duty diesel engine, Energies, 10, 1087, https://doi.org/10.3390/en10081087, 2017.

Manigrasso, M., Vernale, C., and Avino, P.: Traffic aerosol lobar doses deposited in the human respiratory system, Environ. Sci. Pollut. Res. Int., 24, 13866-13873, https://doi.org/10.1007/s11356-015-5666-1, 2017.

Maricq, M. M., and Ning, X.: The effective density and fractal dimension of soot particles from premixed flames and motor vehicle exhaust, J. Aerosol Sci., 35, 1251-1274, https://doi.org/10.1016/j.jaerosci.2004.05.002, 2004.

Maricq, M. M.: Chemical characterization of particulate emissions from diesel engines: A review, J. Aerosol Sci., 38, 1079-1118, https://doi.org/10.1016/j.jaerosci.2007.08.001, 2007.

Martinet, S., Liu, Y., Louis, C., Tassel, P., Perret, P., Chaumond, A., and Andre, M.: Euro 6 Unregulated Pollutant Characterization and Statistical Analysis of After-Treatment Device and Driving-Condition Impact on Recent Passenger-Car Emissions, Environ. Sci. Technol., 51, 5847-5855, https://doi.org/10.1021/acs.est.7b00481, 2017.

Martini, G., Giechaskiel, B., and Dilara, P.: Future European emission standards for vehicles: the importance of the UN-ECE Particle Measurement Programme, Biomarkers, 14 Suppl 1, 29-33, https://doi.org/10.1080/13547500902965393, 2009.

May, A. A., Nguyen, N. T., Presto, A. A., Gordon, T. D., Lipsky, E. M., Karve, M., Gutierrez, A., Robertson, W. H., Zhang, M., Brandow, C., Chang, O., Chen, S. Y., Cicero-Fernandez, P., Dinkins, L., Fuentes, M., Huang, S. M., Ling, R., Long, J., Maddox, C., Massetti, J., McCauley, E., Miguel, A., Na, K., Ong, R., Pang, Y. B., Rieger, P., Sax, T., Truong, T., Vo, T., Chattopadhyay, S., Maldonado, H., Maricq, M. M., and Robinson, A. L.: Gas- and particle-phase primary emissions from in-use, on-road gasoline and diesel vehicles, Atmos. Environ., 88, 247-260, https://doi.org/10.1016/j.atmosenv.2014.01.046, 2014.

Mendoza-Villafuerte, P., Suarez-Bertoa, R., Giechaskiel, B., Riccobono, F., Bulgheroni, C., Astorga, C., and Perujo, A.: NOx, NH3, N2O and PN real driving emissions from a Euro VI heavy-duty vehicle. Impact of regulatory on-road test conditions on emissions, Sci. Total Environ., 609, 546-555, https://doi.org/10.1016/j.scitotenv.2017.07.168, 2017.

Moody, A., and Tate, J.: In Service CO2 and NOX Emissions of Euro 6/VI Cars, Light-and Heavy-dutygoods Vehicles in Real London driving: Taking the Road into the Laboratory, Journal of Earth Sciences and Geotechnical Engineering, 7, 51-62, https://orcid.org/0000-0003-1646-6852, 2017.

Nelson, P. F., Tibbett, A. R., and Day, S. J.: Effects of vehicle type and fuel quality on real-world toxic emissions from diesel vehicles, Atmos. Environ., 42, 5291-5303, https://doi.org/10.1016/j.atmosenv.2008.02.049, 2008.





Pirjola, L., Dittrich, A., Niemi, J. V., Saarikoski, S., Timonen, H., Kuuluvainen, H., Jarvinen, A., Kousa, A., Ronkko, T., and
Hillamo, R.: Physical and Chemical Characterization of Real-World Particle Number and Mass Emissions from City Buses in
Finland, Environ. Sci. Technol., 50, 294-304, https://doi.org/10.1021/acs.est.5b04105, 2016.
Pirjola, L., Niemi, J. V., Saarikoski, S., Aurela, M., Enroth, J., Carbone, S., Saarnio, K., Kuuluvainen, H., Kousa, A., Ronkko,
T., and Hillamo, R.: Physical and chemical characterization of urban winter-time aerosols by mobile measurements in Helsinki,
Finland, Atmos. Environ., 158, 60-75, https://doi.org/10.1016/j.atmosenv.2017.03.028, 2017.
Preble, C., Cados, T., Harley, R., and Kirchstetter, T.: Impacts of Aging Emission Control Systems on In-Use Heavy-Duty
Diesel Truck Emission Rates, AGU Fall Meeting Abstracts, 2017,
Preble, C. V., Dallmann, T. R., Kreisberg, N. M., Hering, S. V., Harley, R. A., and Kirchstetter, T. W.: Effects of Particle
Filters and Selective Catalytic Reduction on Heavy-Duty Diesel Drayage Truck Emissions at the Port of Oakland, Environ.
Sci. Technol., 49, 8864-8871, https://doi.org/10.1021/acs.est.5b01117, 2015.
Preble, C. V., Cados, T. E., Harley, R. A., and Kirchstetter, T. W.: In-Use Performance and Durability of Particle Filters on
Heavy-Duty Diesel Trucks, Environ. Sci. Technol., 52, 11913-11921, https://doi.org/10.1021/acs.est.8b02977, 2018.
Quiros, D. C., Hu, S. H., Hu, S. S., Lee, E. S., Sardar, S., Wang, X. L., Olfert, J. S., Jung, H. J. S., Zhu, Y. F., and Huai, T.:
Particle effective density and mass during steady-state operation of GDI, PFI, and diesel passenger cars, J. Aerosol Sci., 83,
39-54, https://doi.org/10.1016/j.jaerosci.2014.12.004, 2015.
Quiros, D. C., Thiruvengadam, A., Pradhan, S., Besch, M., Thiruvengadam, P., Demirgok, B., Carder, D., Oshinuga, A., Huai,
T., and Hu, S.: Real-world emissions from modern heavy-duty diesel, natural gas, and hybrid diesel trucks operating along
major California freight corridors, Emission Control Science and Technology, 2, 156-172, https://doi.org/10.1007/s40825-
652 016-0044-0, 2016.

Quiros, D. C., Smith, J. D., Ham, W. A., Robertson, W. H., Huai, T., Ayala, A., and Hu, S.: Deriving fuel-based emission
factor thresholds to interpret heavy-duty vehicle roadside plume measurements, J. Air Waste Manag. Assoc., 68, 969-987,
https://doi.org/10.1080/10962247.2018.1460637, 2018.
Rexeis, M., Röck, M., and Hausberger, S.: Comparison of Fuel Consumption and Emissions for Representative Heavy-Duty
Vehicles in Europe, 2018.
Ristimaki, J., Vaaraslahti, K., Lappi, M., and Keskinen, J.: Hydrocarbon condensation in heavy-duty diesel exhaust, Environ.
Sci. Technol., 41, 6397-6402, https://doi.org/10.1021/es0624319, 2007.
Robinson, A. L., Donahue, N. M., Shrivastava, M. K., Weitkamp, E. A., Sage, A. M., Grieshop, A. P., Lane, T. E., Pierce, J.
R., and Pandis, S. N.: Rethinking organic aerosols: semivolatile emissions and photochemical aging, Science, 315, 1259-1262,
https://doi.org/10.1126/science.1133061, 2007.
Rymaniak, L., Ziolkowski, A., and Gallas, D.: Particle number and particulate mass emissions of heavy-duty vehicles in real
operating conditions, MATEC Web of Conferences, 2017, 00025,
Sakurai, H., Park, K., McMurry, P. H., Zarling, D. D., Kittelson, D. B., and Ziemann, P. J.: Size-dependent mixing
characteristics of volatile and nonvolatile components in diesel exhaust aerosols, Environ. Sci. Technol., 37, 5487-5495,
https://doi.org/10.1021/es034362t, 2003a.



Sakurai, H., Tobias, H. J., Park, K., Zarling, D., Docherty, K. S., Kittelson, D. B., McMurry, P. H., and Ziemann, P. J.: On-
line measurements of diesel nanoparticle composition and volatility, Atmos. Environ., 37, 1199-1210,
https://doi.org/10.1016/S1352-2310(02)01017-8, 2003b.
Smit, R., Keramydas, C., Ntziachristos, L., Lo, T. S., Ng, K. L., Wong, H. L. A., and Wong, C.: Evaluation of real-world
gaseous emissions performance of SCR and DPF bus retrofits, Environ. Sci. Technol., https://doi.org/10.1021/acs.est.8b07223,
673 2019.

Tan, Y., Henderick, P., Yoon, S., Herner, J., Montes, T., Boriboonsomsin, K., Johnson, K., Scora, G., Sandez, D., and Durbin,
T. D.: On-Board Sensor-Based NOx Emissions from Heavy-Duty Diesel Vehicles, Environ. Sci. Technol., 53, 5504-5511,
https://doi.org/10.1021/acs.est.8b07048, 2019.
Thiruvengadam, A., Besch, M. C., Carder, D. K., Oshinuga, A., and Gautam, M.: Influence of real-world engine load
conditions on nanoparticle emissions from a DPF and SCR equipped heavy-duty diesel engine, Environ. Sci. Technol., 46,
1907-1913, https://doi.org/10.1021/es203079n, 2012.
Thiruvengadam, A., Besch, M. C., Thiruvengadam, P., Pradhan, S., Carder, D., Kappanna, H., Gautam, M., Oshinuga, A.,
Hogo, H., and Miyasato, M.: Emission rates of regulated pollutants from current technology heavy-duty diesel and natural gas
goods movement vehicles, Environ. Sci. Technol., 49, 5236-5244, https://doi.org/10.1021/acs.est.5b00943, 2015.
Vaaraslahti, K., Virtanen, A., Ristimaki, J., and Keskinen, J.: Nucleation mode formation in heavy-duty diesel exhaust with
and without a particulate filter, Environ. Sci. Technol., 38, 4884-4890, https://doi.org/10.1021/es0353255, 2004.
Van Setten, B. A., Makkee, M., and Moulijn, J. A.: Science and technology of catalytic diesel particulate filters, Catalysis
Reviews, 43, 489-564, https://doi.org/10.1081/CR-120001810, 2001.
Vojtisek-Lom, M., Pechout, M., Dittrich, L., Beranek, V., Kotek, M., Schwarz, J., Vodicka, P., Milcova, A., Rossnerova, A.,
Ambroz, A., and Topinka, J.: Polycyclic aromatic hydrocarbons (PAH) and their genotoxicity in exhaust emissions from a
diesel engine during extended low-load operation on diesel and biodiesel fuels, Atmos. Environ., 109, 9-18,
https://doi.org/10.1016/j.atmosenv.2015.02.077, 2015.
Wang, T., Quiros, D. C., Thiruvengadam, A., Pradhan, S., Hu, S., Huai, T., Lee, E. S., and Zhu, Y.: Total Particle Number
Emissions from Modern Diesel, Natural Gas, and Hybrid Heavy-Duty Vehicles During On-Road Operation, Environ. Sci.
Technol., 51, 6990-6998, https://doi.org/10.1021/acs.est.6b06464, 2017.
Watne, A. K., Psichoudaki, M., Ljungstrom, E., Le Breton, M., Hallquist, M., Jerksjo, M., Fallgren, H., Jutterstrom, S., and
Hallquist, A. M.: Fresh and Oxidized Emissions from In-Use Transit Buses Running on Diesel, Biodiesel, and CNG, Environ.
Sci. Technol., 52, 7720-7728, https://doi.org/10.1021/acs.est.8b01394, 2018.
Williams, M., and Minjares, R.: A technical summary of Euro 6/VI vehicle emission standards, International Council for Clean
Transportation (ICCT), Washington, DC, accessed July, 10, 2017, 2016.
Zavala, M., Molina, L. T., Yacovitch, T. I., Fortner, E. C., Roscioli, J. R., Floerchinger, C., Herndon, S. C., Kolb, C. E.,
Knighton, W. B., Paramo, V. H., Zirath, S., Mejia, J. A., and Jazcilevich, A.: Emission factors of black carbon and co-pollutants
from diesel vehicles in Mexico City, Atmos. Chem. Phys., 17, 15293-15305, https://doi.org/10.5194/acp-17-15293-2017,
702 2017.

Zhang, S. J., Wu, Y., Liu, H., Huang, R. K., Yang, L. H. Z., Li, Z. H., Fu, L. X., and Hao, J. M.: Real-world fuel consumption
and CO2 emissions of urban public buses in Beijing, Appl. Energ., 113, 1645-1655,
https://doi.org/10.1016/j.apenergy.2013.09.017, 2014.





Zheng, Z., Johnson, K. C., Liu, Z., Durbin, T. D., Hu, S., Huai, T., Kittelson, D. B., and Jung, H. S.: Investigation of solid
particle number measurement: Existence and nature of sub-23 nm particles under PMP methodology, J. Aerosol Sci., 42, 883-
897, https://doi.org/10.1016/j.jaerosci.2011.08.003, 2011.
Zheng, Z., Durbin, T. D., Xue, J., Johnson, K. C., Li, Y., Hu, S., Huai, T., Ayala, A., Kittelson, D. B., and Jung, H. S.:
Comparison of particle mass and solid particle number (SPN) emissions from a heavy-duty diesel vehicle under on-road driving
conditions and a standard testing cycle, Environ. Sci. Technol., 48, 1779-1786, https://doi.org/10.1021/es403578b, 2014.
Zheng, Z. Q., Durbin, T. D., Karavalakis, G., Johnson, K. C., Chaudhary, A., Cocker, D. R., Herner, J. D., Robertson, W. H.,
Huai, T., Ayala, A., Kittelson, D. B., and Jung, H. S.: Nature of Sub-23-nm Particles Downstream of the European Particle
Measurement Programme (PMP)-Compliant System: A Real-Time Data Perspective, Aerosol Sci. Tech., 46, 886-896,
https://doi.org/10.1080/02786826.2012.679167, 2012.
Zhu, Y. F., Hinds, W. C., Kim, S., Shen, S., and Sioutas, C.: Study of ultrafine particles near a major highway with heavy-duty
diesel traffic, Atmos. Environ., 36, 4323-4335, https://doi.org/10.1016/S1352-2310(02)00354-0, 2002.



**Table 1.** Comparison of the average emission data[a] for PM and PN from the present study with literature data.

| PM/PN | | | | | | |
|---|---|---|---|---|---|---|
| Vehicle type | Speed km h$^{-1}$ | Dp range nm | Method | Instrument | EF$_{PM}$ mg (kg fuel)$^{-1}$ | EF$_{PN}$ # (kg fuel)$^{-1}$ 10$^{14}$ |
| **Euro III HDT in this study** | 26±6[b] | 5.6-560 | roadside | EEPS | 684±365 | 20.3±11.7 |
| Euro III bus (Hallquist et al., 2013) | acceleration | 5.6-560 | roadside | EEPS | 6.7-2074 | 0.11-45 |
| | constant speed | 5.6-560 | roadside | EEPS | 151-273 | 0.12-4.2 |
| Euro III bus with DPF (Hallquist et al., 2013) | acceleration | 5.6-560 | roadside | EEPS | 62-2465 | 1.9-23 |
| | constant speed | 5.6-560 | roadside | EEPS | 41-142 | 1.1-9.7 |
| Euro III bus (Pirjola et al., 2016) | ≤25 (bus depot) | PM$_1$ D$_p$ ≥2.5 | plume chasing | ELPI[c] CPC | 1240±220[b] | 20.6±3.2[b] |
| | ≤45 (bus line) | PM$_1$ D$_p$ ≥2.5 | plume chasing | ELPI[c] CPC | 500 | 17.7 |
| Euro III bus with DPF+SCR (Watne et al., 2018) | acceleration | 5.6-560 | roadside | EEPS | 8.9±0.2 | 0.12±0.12 |
| Euro III bus with DPF+SCR (Liu et al., 2019) | stop and go (bus stop) | 5.6-560 | roadside | EEPS | 30±26[b] | 14.0±3.0[b] |
| Euro III diesel bus and truck (Zavala et al., 2017) | driving cycle | 35-1000 | plume chasing & roadside | SP-AMS[f] | 4300 | - |
| **Euro IV HDT in this study** | 23±8[b] | 5.6-560 | roadside | EEPS | 172±68 | 8.7±3.0 |
| Euro IV bus with EGR (Hallquist et al., 2013) | acceleration | 5.6-560 | roadside | EEPS | 562-3089 | 13-44 |
| | constant speed | 5.6-560 | roadside | EEPS | 91-489 | 5.8-47 |
| Euro IV bus with EGR+DPF (Hallquist et al., 2013) | acceleration | 5.6-560 | roadside | EEPS | 177-650 | 5.1-13 |
| | constant speed | 5.6-560 | roadside | EEPS | 58-61 | 2.6-3.1 |
| Euro IV bus with EGR+DPF (Pirjola et al., 2016) | ≤25 (bus depot) | PM$_1$ D$_p$ ≥2.5 | plume chasing | ELPI[c] CPC | 1190±520[b] | 8.9±1.6[b] |
| Euro IV bus with SCR (Watne et al., 2018) | acceleration | 5.6-560 | roadside | EEPS | 145-560 | 3-13 |
| Euro IV diesel bus and truck (Zavala et al., 2017) | driving cycle | 35-1000 | plume chasing and roadside | SP-AMS[f] | 1800 | - |
| **Euro V HDT in this study** | 27±7[b] | 5.6-560 | roadside | EEPS | 146±49 | 9.7±2.7 |
| Euro V bus+SCR (Hallquist et al., 2013) | acceleration | 5.6-560 | roadside | EEPS | 125-766 | 4.4-92 |
| | constant speed | 5.6-560 | roadside | EEPS | 41-509 | 2.7-33 |
| Euro V bus (Watne et al., 2018) | acceleration | 5.6-560 | roadside | EEPS | 145±70 | 3.0±1.7 |
| Euro V HDV with SCR (Rymaniak et al., 2017) | average at 45 | PM/ 5.6-560 | PEMS | MSS[e] EEPS | 1840 [d] | 0.09 [d] |
| Euro V bus with SCR (Liu et al., 2019) | stop and go (bus stop) | 5.6-560 | roadside | EEPS | 180±15[b] | 6.5±2.9[b] |
| Euro V diesel bus and truck (Zavala et al., 2017) | driving cycle | 35-1000 | plume chasing and roadside | SP-AMS[f] | 720 | - |
| **EEV HDT in this study** | 25±8[b] | 5.6-560 | roadside | EEPS | 78±35 | 16.5±23.6 |





| | | | | | | |
|---|---|---|---|---|---|---|
| EEV bus with EGR +DPF (Pirjola et al., 2016) | ≤25 (bus depot) | $PM_1$/ $D_p$ ≥2.5 | plume chasing | ELPI[c] CPC | 400±280[b] | 2.1±0.1[b] |
| EEV bus with SCR (Pirjola et al., 2016) | ≤25 (bus depot) | $PM_1$/ $D_p$ ≥2.5 | plume chasing | ELPI[c] CPC | 280±170[b] | 7.0±3.8[b] |
| EEV with DOC+DPF+SCR (Rymaniak et al., 2017) | average at 45 | PM/ 5.6-560 | PEMS | MSS[e] EEPS | 236[d] | 0.02[d] |
| EEV bus (Jarvinen et al., 2019) | stop and go | $PM_1$/ $D_p$ ≥3 | plume chasing | ELPI[c] CPC | 200 | 8.6 |
| **Euro VI HDT in this study** | 29±8[b] | 5.6-560 | roadside | EEPS | 5±2 | 8.5±4.6 |
| Euro VI bus (Jarvinen et al., 2019) | stop and go | $PM_1$/ $D_p$ ≥3 | plume chasing | ELPI[c] CPC | 70 | 5 |
| Euro VI HDGV (Moody and Tate, 2017) | 13-86 | - | PEMS | - | 28-33[d] | - |
| Euro VI HDT (Grigoratos et al., 2019) | 65-74 | - | PEMS | - | - | 0.002-0.01[d] |
| HDV with DPF (Wang et al., 2017; Quiros et al., 2016) | 13-80 | PM $D_p$ ≥5 | PEMS | gravimetric CPC | 12-41[d] | 0.006-13.2 |
| Heavy-duty HDV with DPF+SCR (Thiruvengadam et al., 2015) | driving cycle | PM | chassis dynamometer | gravimetric | 6-29[d] | - |
| HDV with DPF+SCR (Jiang et al., 2018) | driving cycle | $PM_{2.5}$ | chassis dynamometer | gravimetric | 3-97[d] | - |
| HDT (model year 2004- 2006) (Preble et al., 2015) | accelerating or cruise at 48 | $D_p$ ≥2.5 | roadside | CPC | - | 47.2±9.7 |
| HDT with SCR+DPF (model year 2010- 2013) (Preble et al., 2015) | | $D_p$ ≥2.5 | roadside | CPC | - | 15.9±11.5 |
| HDV (mean model year 2005) (Bishop et al., 2015) | 15.7-16.8 | $PM_{1.2}$ | OHMS[g] | digital mass monitor | 650 | - |
| HDV (mean model year 2009) (Bishop et al., 2015) | 7.7-9.3 | $PM_{1.2}$ | OHMS[g] | digital mass monitor | 31 | - |
| HDV without after-treatment (Quiros et al., 2018) | driving cycle | $PM_{2.5}$ | chassis dynamometer | gravimetric | 1980[d] | - |
| HDV+DPF (Quiros et al., 2018) | driving cycle | $PM_{2.5}$ | chassis dynamometer | gravimetric | 6-9[d] | - |
| **HDT without available Euro type information** | 27±7[b] | 5.6-560 | roadside | EEPS | 47±23 | 7.5±7.3 |
| **Total Swedish HDT** | 28±7[b] | 5.6-560 | roadside | EEPS | 96±36 | 9.6±2.7 |
| **Total non-Swedish HDT** | 26±8[b] | 5.6-560 | roadside | EEPS | 117±42 | 11.1±4.2 |

[a] Given errors are at 95% CI.
[b] Standard deviation.
[c] ELPI, Electrical Low-Pressure Impactor.
[d] Average fuel consumption of 0.26 L km[-1] for HDV during long haul and regional delivery tests (Rexeis et al., 2018), the
density of 0.815 kg dm[-3] (Swedish Environmental Protection Agency, 2013) of diesel particles were assumed for unit
conversion.
[e] MSS, Micro Soot Sensor.



[f] SP-AMS, Soot Particle Aerosol Mass Spectrometer.
[g] OHMS, On-Road Heavy-Duty Vehicle Emissions Monitoring System.



**Table 2.** Comparison of the average emission data[a] for $NO_x$, $NO_2/NO_x$, CO and HC from the present study with literature data.

| Vehicle type | Speed km h$^{-1}$ | Method | $EF_{NOx}$[b] g (kg fuel)$^{-1}$ | $EF_{NO2}/EF_{NOx}$[b] mass ratio % | $EF_{CO}$[c] g (kg fuel)$^{-1}$ | $EF_{HC}$[c] g (kg fuel)$^{-1}$ |
|---|---|---|---|---|---|---|
| **Euro III HDT in this study** | 26±6[d] | roadside | 43.3±31.5 | 7.5±4.1 | 36.0±13.2 | 0.8±1.3 |
| Euro III bus (Hallquist et al., 2013) | acceleration | roadside | 16.1±9.7 | - | 16.1±16.1 | <13 |
| Euro III bus (Pirjola et al., 2016) | ≤25 (bus depot) | plume chasing | 12.7±1.8[d] | - | - | - |
| | ≤45 (bus line) | plume chasing | 20.5 | - | - | - |
| Euro III bus with DPF+SCR (Watne et al., 2018) | acceleration | roadside | - | - | 13±10 | 0.02 |
| Euro III HDV (Lau et al., 2015) | 64 ± 13[d] | plume chasing | - | 24±4 | - | - |
| Euro III &IV HDV (Kousoulidou et al., 2008) | - | model | - | 14 | - | - |
| Euro III HGV (Carslaw et al., 2011) | Average at 31 | roadside | 16.2±1.0[f] | - | - | - |
| **Euro IV HDT in this study** | 23±8[d] | roadside | 19.8±10.1 | 2.7±2.9 | 22.1±10.3 | 0.7±1.1 |
| Euro IV bus (Hallquist et al., 2013) | acceleration | roadside | 12.9±6.5 | - | 16.1±16.1 | <13 |
| Euro IV bus with EGR+DPF (Pirjola et al., 2016) | ≤25 (bus depot) | plume chasing | 23.4±6.1[d] | - | - | - |
| Euro IV bus with SCR (Watne et al., 2018) | roadside | acceleration | - | - | 220-230 | 0.3-0.6 |
| Euro IV HDV (Lau et al., 2015) | 64 ± 13[d] | plume chasing | - | 28±5 | - | - |
| Euro IV HGV (Carslaw et al., 2011) | average at 31 | roadside | 10.3±1.4[f] | - | - | - |
| **Euro V HDT in this study** | 27±7[d] | roadside | 22.2±3.8 | 6.0±2.8 | 22.8±5.1 | 0.9±0.4 |
| Euro V bus (Hallquist et al., 2013) | acceleration | roadside | 35.5±9.7 | - | 9.7±3.2 | <13 |
| Euro V bus with SCR (Liu et al., 2019) | stop and go (bus stop) | roadside | 9.8±3.5[d] | 3.7±1.5[d] | 28[e] | 2.2[e] |
| Euro V HDV (Lau et al., 2015) | 64 ± 13[d] | plume chasing | - | 40±14 | - | - |
| Euro V HDV (Kousoulidou et al., 2008) | - | model | - | 18 | - | - |
| Euro V HGV (Carslaw et al., 2011) | average at 31 | roadside | 13.3±5.8[f] | - | - | - |
| **EEV HDT in this study** | 25±8[d] | roadside | 13.6±6.7 | 6.3±3.7 | 18.0±10.1 | 0.2±0.4 |
| EEV bus with EGR +DPF (Pirjola et al., 2016) | ≤25 (bus depot) | plume chasing | 32.9±7.6[d] | - | - | - |
| EEV bus with SCR (Pirjola et al., 2016) | ≤25 (bus depot) | plume chasing | 39.8±4.2[d] | - | - | - |
| **Euro VI HDT in this study** | 29±8[d] | roadside | 3.1±1.0 | 22.5±4.2 | 15.5±2.2 | 1.0±0.5 |





| | | | | | | |
|---|---|---|---|---|---|---|
| Euro VI HDT (Grigoratos et al., 2019) | 65-74 | PEMS | 0.3-31.3 | - | 2.8-22.3 | 0.3-3.1 |
| Euro VI HDV (Kousoulidou et al., 2008) | - | model | - | 35 | - | - |
| Euro VI HDV (Moody and Tate, 2017) | driving cycle | PEMS | 2.2[f] | - | - | - |
| Heavy-duty HDV with DPF+SCR (Thiruvengadam et al., 2015) | driving cycle | chassis dynamometer | 3.8-27.8[f] | - | 0.1-13.4[f] | <0.64[f] |
| HDV with DPF+SCR (Jiang et al., 2018) | driving cycle | chassis dynamometer | 0.2-66.4[f] | - | 0.006-14.9[f] | <1.3[f] |
| HDV with DOC+DPF+SCR (Quiros et al., 2016) | 12.7-85.6 | mobile laboratory | 1.7-11.8[f] | - | 0.9-2.8[f] | 0.1-0.4[f] |
| HDV (May et al., 2014) | driving cycle | chassis dynamometer | 30-43 | - | - | - |
| HDV with SCR (May et al., 2014) | driving cycle | chassis dynamometer | 11 | - | - | - |
| HDV fleet average (Haugen et al., 2018) | 22.5±0.9 | remote sensing | 12.4±0.6 | 8.9 | 5.9±0.9 | 2.2±0.4 |
| HDT (model year 2004- 2006) (Preble et al., 2015) | accelerating or cruise at 48 | roadside | 16.5±1.7 | 3.4±1.8 | - | - |
| HDT with SCR+DPF (model year 2010- 2013) (Preble et al., 2015) | | roadside | 5.1±1.2 | 22.1±8.4 | - | - |
| HDT (model year 2001) (Burgard et al., 2006) | 5-25 | roadside | - | 9.1±0.5 | 26.0±2.1 | 1.8±0.6 |
| HDT (model year 2000) (Burgard et al., 2006) | 20-40 | roadside | - | 6.1±0.1 | 37.9±1.6 | 3.3±0.4 |
| Fleet average in 2006 (Bishop and Stedman, 2008) | 28-36 | roadside | 2-5 | - | 17-24 | 1.9-2.3 |
| HDT fleet average (Dallmann et al., 2012) | 65 | roadside | 28±1.5 | 7.0 | 8.0±1.2 | - |
| HDT (mean model year 2004) (Bishop et al., 2013) | 22.2±0.4 | remote sensing | 20.6±0.6[d] | 9.7 | 8.2±0.6[d] | 3.7±0.1[d] |
| HDT (mean model year 2009) (Bishop et al., 2013) | 7.8±0.1 | remote sensing | 19.9±0.3[d] | 9.0 | 7.3±0.5[d] | 0.6±0.6[d] |
| **HDT without available Euro type information** | 27±7[d] | roadside | 7.8±4.5 | 13.9±6.3 | 20.7±5.6 | 0.8±0.6 |
| **Total Swedish HDT** | 28±7[d] | roadside | 10.7±1.8 | 15.9±2.5 | 18.6±1.9 | 0.9±0.3 |
| **Total non-Swedish HDT** | 26±8[d] | roadside | 13.0±2.5 | 12.7±3.0 | 19.1±3.0 | 0.9±0.6 |

[a] Given errors are at 95% CI.
[b] In $NO_2$ equivalents.
[c] RSD data. For the RSD data sets of multiple individuals, negative values were replaced by zero when calculating the averages.
[d] Standard deviation.
[e] Median.
[f] Average fuel consumption of 0.26 L km$^{-1}$ for HDV during long haul and regional delivery tests (Rexeis et al., 2018), the
density of 0.815 kg dm$^{-3}$ (Swedish Environmental Protection Agency, 2013) of diesel particles were assumed.



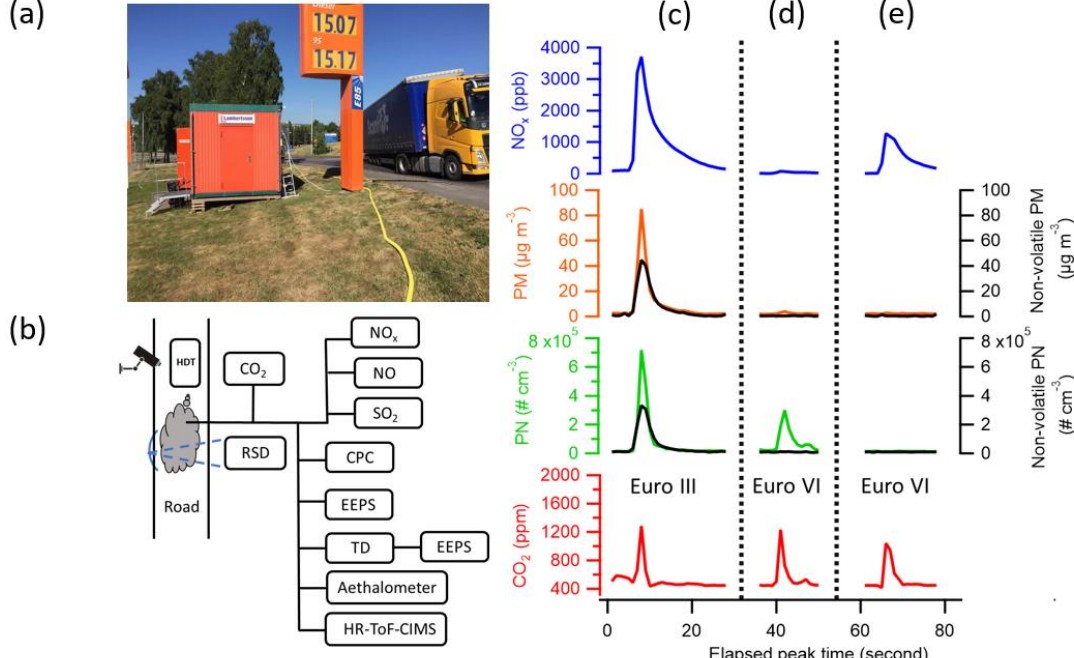


**Figure 1.** (a) Sampling site at the roadside in Gothenburg, Sweden, (b) schematic of the experimental set-up. HDT (Heavy-duty truck), RSD (Remote Sensing Device), CPC (Condensation Particle Counter), EEPS (Engine Exhaust Particle Sizer Spectrometer), TD (Thermodenuder) and HR-ToF-CIMS* (High-Resolution Time-of-Flight Chemical Ionization Mass Spectrometer) and examples of signals from three passing HDTs. Concentrations of $CO_2$, PN, non-volatile PN, PM, non-volatile PM, and $NO_x$ from (c) a typical Euro III HDTs and (d) a typical Euro VI HDTs and (e) a Euro VI HDTs with low PN emission. *The data from the HR-ToF-CIMS will be presented elsewhere.



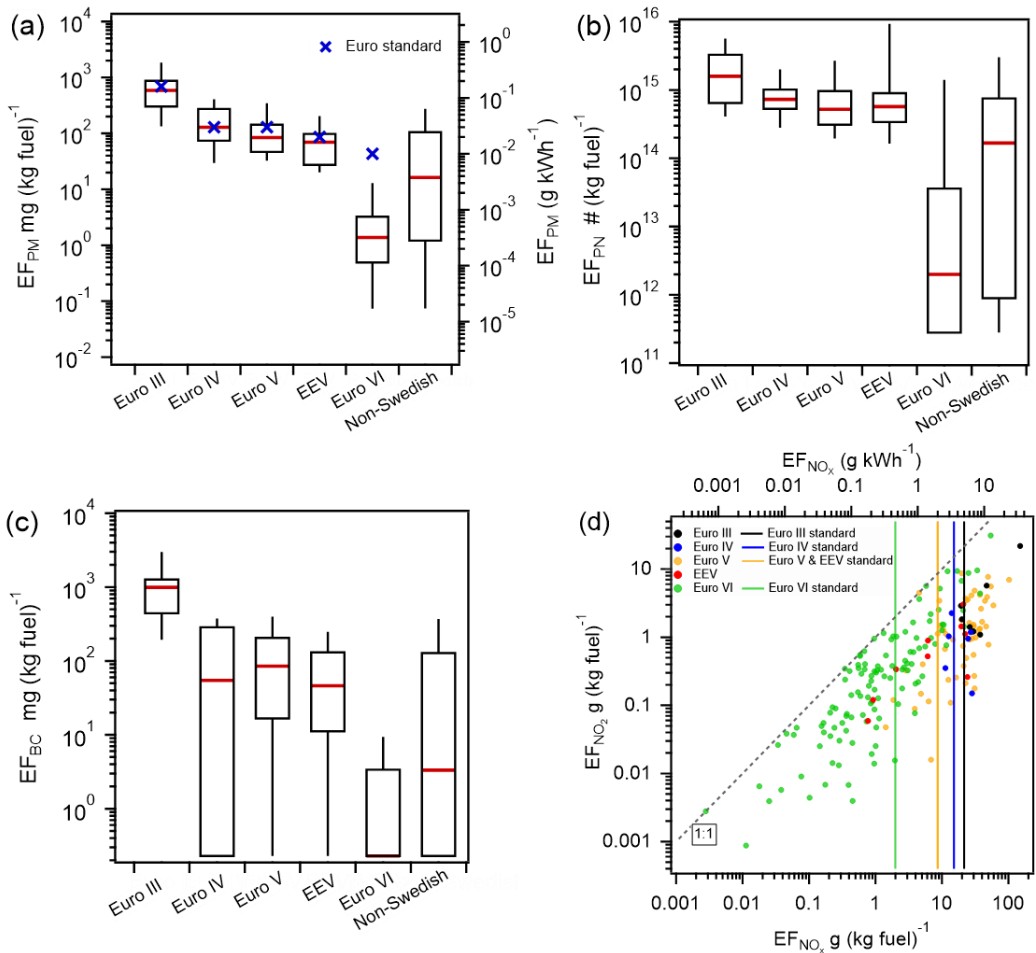

**Figure 2.** (a) $EF_{PM}$, (b) $EF_{PN}$, (c) $EF_{BC}$, (d) $EF_{NO2}$ and $EF_{NOx}$ for Euro III to Euro VI and non-Swedish HDTs. Non-detectable pollutant emission signals for captured plumes have been replaced by $EF_{min}$. For box-and-whisker plots, the top and the bottom line of the box are 75th and 25th percentiles of the data, the red line inside the box is the median, and the top and bottom whiskers are 90th and 10th percentiles. $EF_{NOx}$ in (d) are in $NO_2$ equivalents. Note that the comparison with the emission standard is only indicative as they are based on test cycle performance.

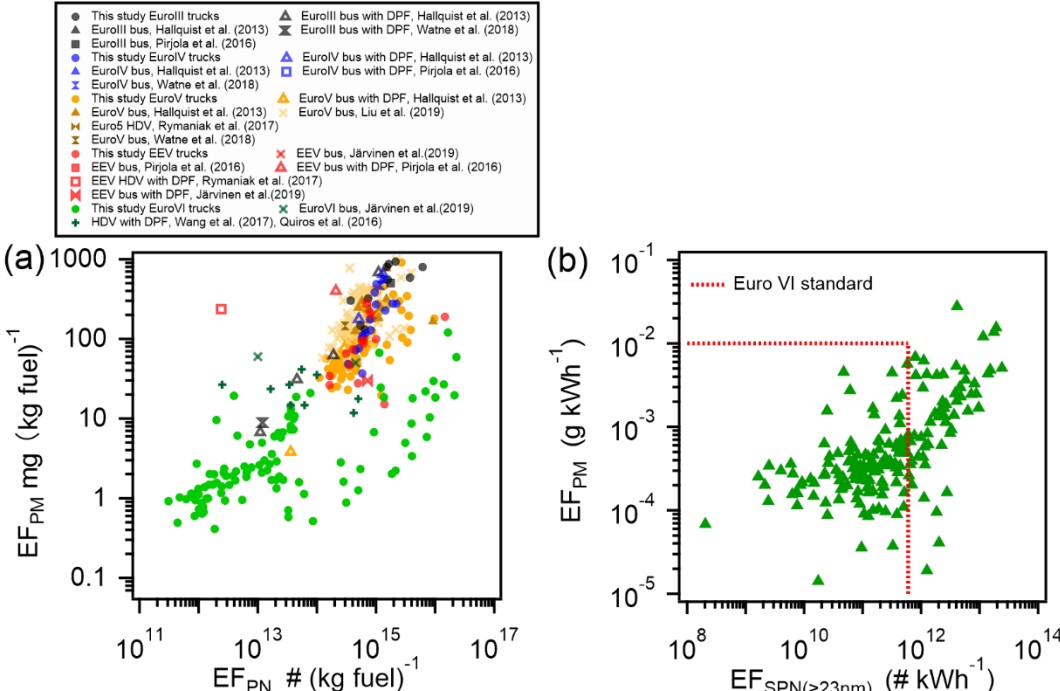

**Figure 3.** (a) $EF_{PM}$ and $EF_{PN}$ of individual HDTs in this study and previous studies and (b) the relationship between $EF_{PM}$ and $EF_{SPN}$ of Euro VI HDTs. Red dashed lines represent Euro emission standards (horizontal: PM emission standard; vertical: SPN emission standard). Note that the comparison with the emission standard is only indicative as they are based on test cycle performance.



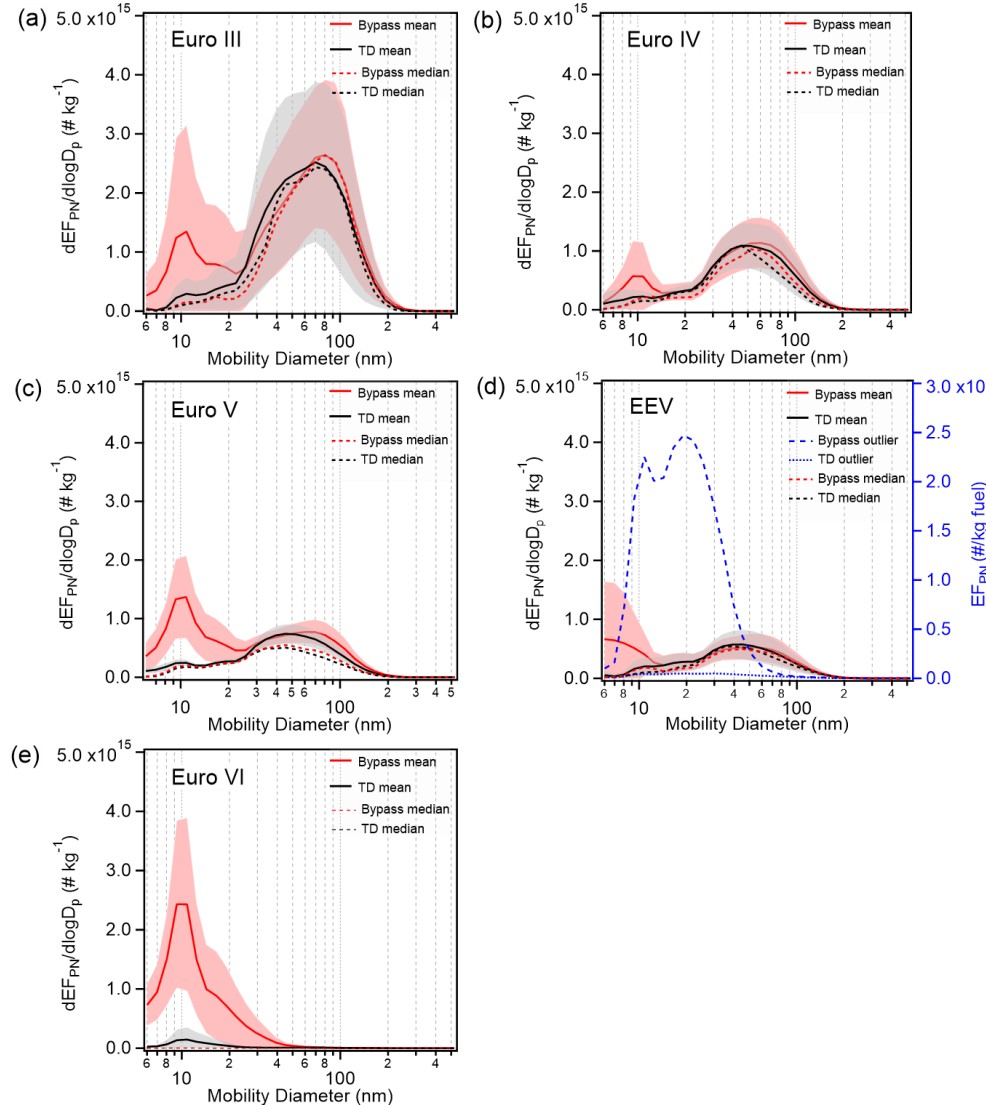

**Figure 4.** Mean and median size-resolved $EF_{PN}$ and $EF_{non\text{-}volatile\ PN}$ of different Euro class HDTs. Shaded regions in (a-e) represent the statistical 95% confidence interval. One HDT with extremely different EF in (d) was excluded and shown in blue.



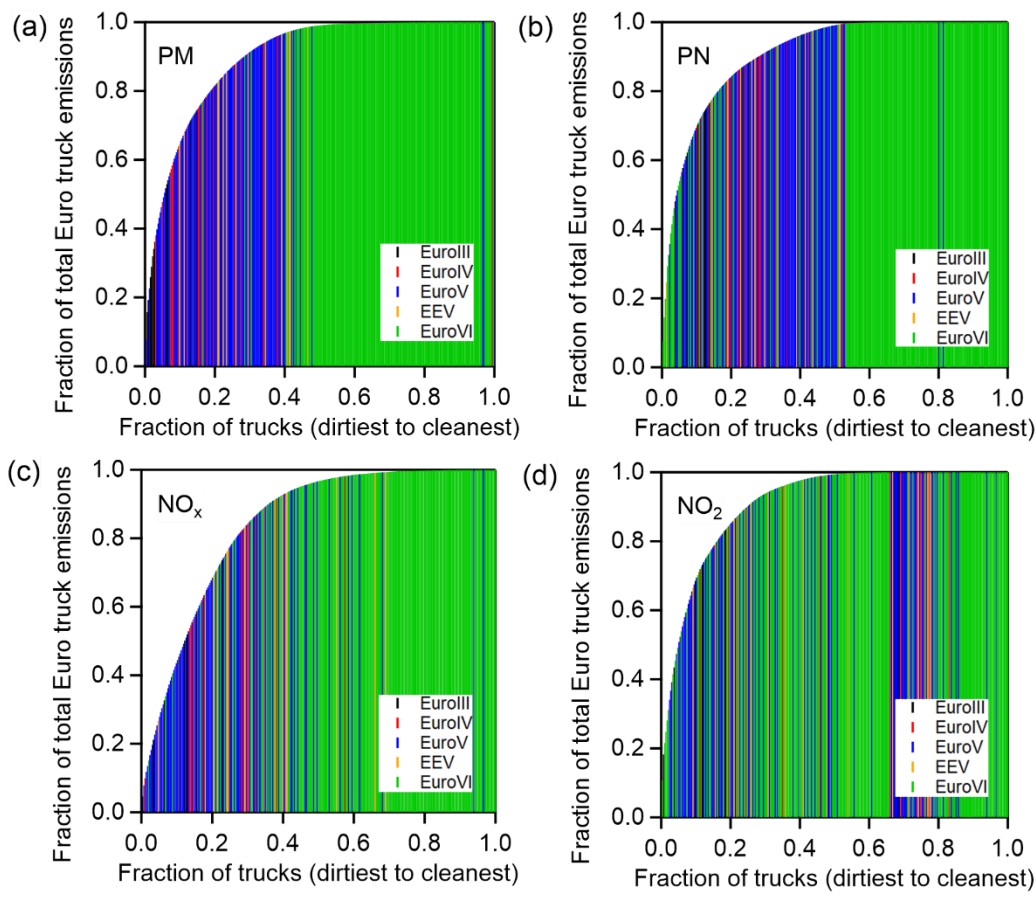

**Figure 5.** Cumulative emission factor distribution for (a) PM, (b) PN, (c) NO$_x$, and (d) NO$_2$ measured in HDT exhaust plumes with HDTs ranked from the highest to lowest in terms of emission factors.

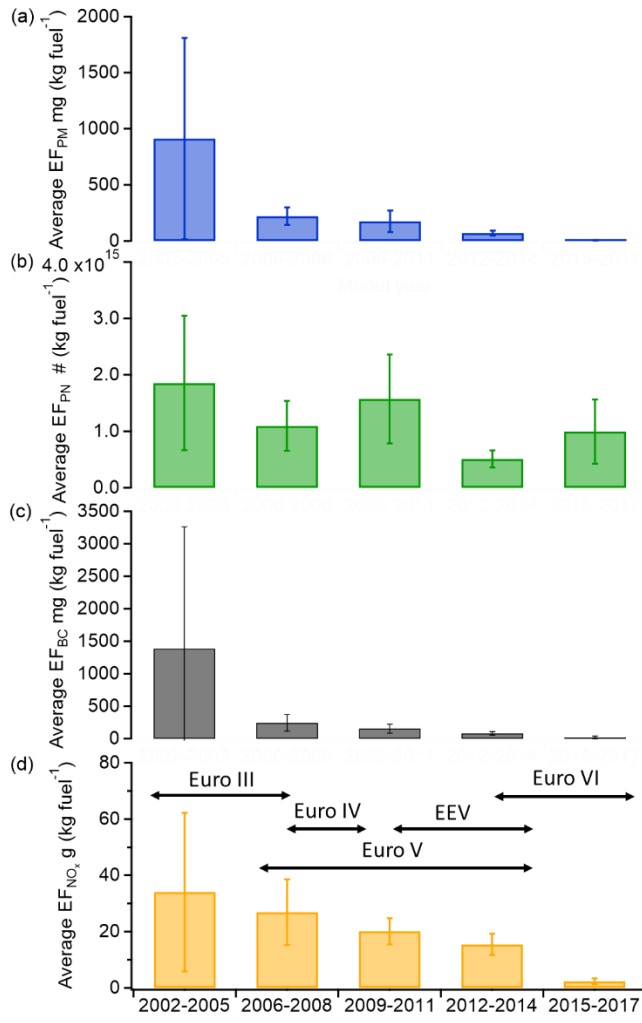

**Figure 6.** Changes of average EFs of (a) PM, (b) PN, (c) BC and (d) NO$_x$ with respect to the registration year of HDTs. Error bars represent the statistical 95% confidence interval. Black arrows mark the years that the particular type of HDT examined in this study was first registered.



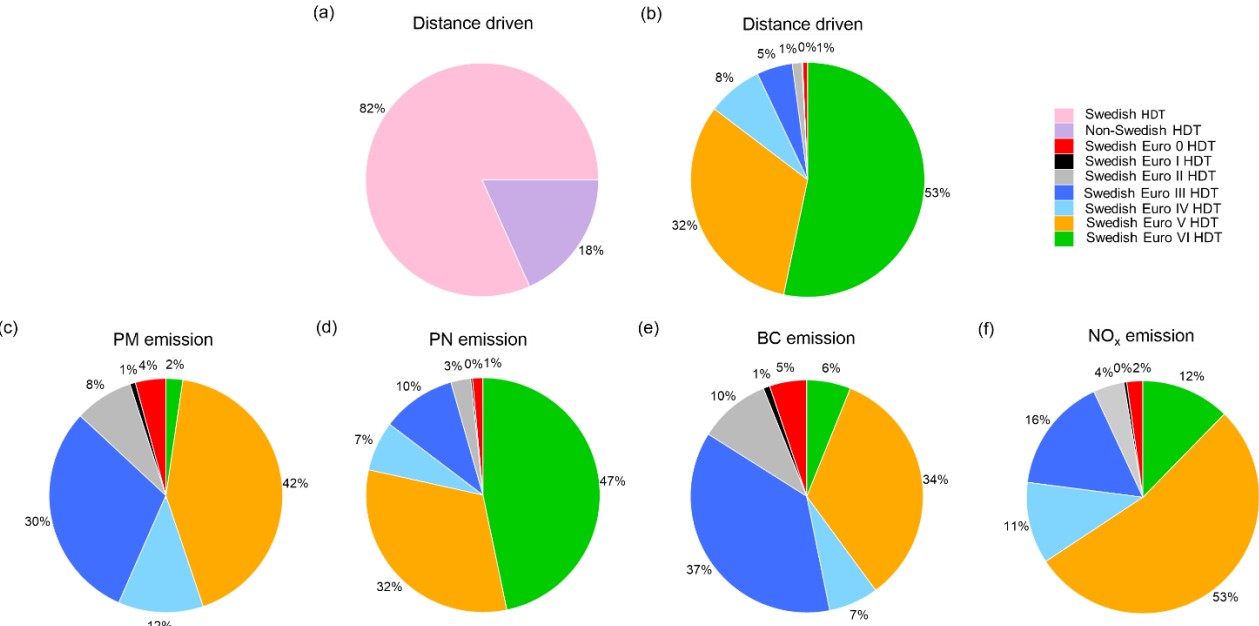

766

**Figure 7.** Relative contributions of kilometers driven by (a) Swedish and Non-Swedish HDTs and (b) Swedish Euro 0, Euro I, Euro II, Euro III, Euro IV, Euro V and Euro VI HDTs on Swedish roads during 2018. Approximation of contributions of pollutants emitted from Swedish HDTs in each Euro class to the total (c) PM, (d) PN, (e) BC and (f) NO$_x$ emissions.