# Peer review of "A transition in atmospheric emissions of particles and gases from on-road heavy-duty trucks"

_Atmospheric Chemistry and Physics, 2019_

## Referee Comment (RC1) · Anonymous Referee #1 · 5 Nov 2019

General comments

This paper uses fast roadside measurements of a range of gaseous and particulate emissions for heavy duty vehicles in Sweden. A relatively large sample size of measurements is used to infer the emission characteristics of different vehicle types - mostly by Euro status. The paper provides an up to date understanding of the evolution of emissions of important species that include non-regulatory species. The paper is generally well-written but perhaps lacks a clear explanation of what the new findings are and how they differ from previous work. Indeed, the size of Table 1 and 2 indicate that a considerable amount of work has been carried out before in this area. Nevertheless,

emissions from road vehicles continually evolve and on balance I think this study does contribute some useful, up to date emissions from an important class of vehicles.

Specific comments

Line 254 where it is stated non-Swedish trucks tend to have higher emissions. I think this statement needs to be more robust. As the authors point out, there is no Euro class information for non-Swedish trucks (and therefore no knowledge of the numbers in each Euro class). Given the large range of emissions by Euro status, I don't think this statement is particularly defensible. Furthermore, all the data shown in Table 1/2 I think have overlapping 95% CI when making this comparison. Also, the statement "which may be attributed to the more stringent domestic goals regarding pollution, clean air, greenhouse gas emissions, energy efficiency, and innovative sustainable solutions..." is vague. Is it the case that Swedish annual technical inspections are more stringent than other countries? I doubt there is evidence for that.

How sure can the authors be that certain Euro classes of vehicles have certain technologies fitted? For example, is it known that any of the tested vehicles were retrofitted in some way?

Where comments are made about decreases (or changes in general) I think it is important to provide the corresponding uncertainty. It seems that in many cases that there will be overlapping confidence intervals and therefore important to convey where differences are statistically clear or not.

How was the sigma of the background component of $CO_2$ calculated? This was not based on upwind measurements, right? Presumably the variation in background cannot be represented by a single value and it varies also. Some more details are needed. It would be helpful to have a Figure that shows a 'typical' peak being analysed showing the concentration of $CO_2$ and all other species. This would also help demonstrate that a measurement frequency of ~1 Hz is sufficient to capture an individual vehicle plume. Moreover, a discussion on the effect of sampling rate (and potentially different sampling

rates) on the on the extracted plume characteristics / metrics would be beneficial.

Do the authors have any information about the engine type or manufacturer to understand whether that is an important factor that could explain some of the differences observed? Earlier on in the text it is stated that these types of factor can be important in determining emissions, so it would b euseful to explain this. Similarly, is there any difference in the size of vehicle sampled (e.g. by engine size or kerb weight)?

Section 4 (Atmospheric implications and conclusions) does not actually consider the atmospheric implications. I think it should - and if it did - it would strengthen the paper. For example, it would be useful to consider the implications for near-road exposures, and consider how UFP could evolve through coagulation etc. as plumes disperse away from roads. Reducing PM mass is clearly important but if the consequence in doing so increase PN, then that could be important.

---

## Referee Comment (RC2) · Anonymous Referee #2 · 8 Nov 2019

This study reports roadside measurements of 556 heavy-duty trucks in Sweden. The paper uses these measurements to investigate how several pollutants of interest vary by each truck's Euro pollutant emissions standard, and carries out several additional analyses including the skewness of the distribution of emitters. In addition, the study does a very extensive comparison to past relevant literature. I think the paper is a valuable contribution to the literature and should be published after considering the comments below.

The comments below that I think require the most attention have to do with adding additional analysis or at least discussion on how various amounts of exhaust dilution

[Figure]

impact your particle number emission factors. This includes both variability in dilution among different trucks in your measurements, and especially differences between your study and the past work used for comparison.

Comments:

Line 104: Please consider adding a short statement here related to how the uphill conditions could impact results. You discuss this later in the paper, but you might want to alert the reader to this, and point to where you discuss it.

Line 163: Please discuss how you chose t1 and t2 for the integration. And more importantly, how did you deal with the different plume widths for NOx versus other pollutants? How sensitive are results to chosen t1 and t2?

Line 188-200: See major comments above. PN measurements would be highly dependent on the amount of dilution the plume has undergone between the engine and the measurement. I would imagine this would contribute to differences between your measured emission factors and emission standards. What are the dilution requirements when certifying for Euro standards?

Line 200-202: Could variability in dilution contribute to the scatter too? Please think this through for all sections that discuss PN emissions results.

Line 240-241: Please make sure to include text in figure captions when you are not including data from all trucks. Are you sure that leaving these data out doesn't lead to a problem with biasing the results? I would imagine that if you are not including results for trucks that have measured concentrations below measurement detection limits, you'd be leaving out the cleanest trucks (though could also be due to the plume missing the sample line). Please think this through for all sections that report results that remove trucks with measurements below detection limits.

Table 1: I don't understand how you've categorized this table. For example, I see studies in this table that are not performed in Europe but are under the Euro VI category. Also, I noticed papers that you are citing in the study and that have emission factor results, but are not in this table. Please ensure you have considered all relevant studies.

Figure 2: For Euro III, it seems that the EF for black carbon is higher than for PM. How could this be?

Figure 4: This is very interesting. You might consider comparing these size resolved emission factors to previous studies that report similar EFs.

---

## Author Comment (AC1) · 18 Dec 2019

General comments

**Author Response:** We appreciate the careful review and constructive suggestions. Our point-by-point responses to the reviewer's general and specific comments are presented below. The changes to the initial manuscript text and supplement illustrations are marked in red.

*This paper uses fast roadside measurements of a range of gaseous and particulate emissions for heavy-duty vehicles in Sweden. A relatively large sample size of measurements is used to infer the emission characteristics of different vehicle types-mostly by Euro status. The paper provides an up to date understanding of the evolution of emissions of important species that include non-regulatory species. The paper is generally well-written but perhaps lacks a clear explanation of what the new findings are and how they differ from previous work. Indeed, the size of Table 1 and 2 indicate that a considerable amount of work has been carried out before in this area. Nevertheless, emissions from road vehicles continually evolve and on balance I think this study does contribute some useful, up to date emissions from an important class of vehicles.*

*1. Line 254 where it is stated non-Swedish trucks tend to have higher emissions. I think this statement needs to be more robust. As the authors point out, there is no Euro class information for non-Swedish trucks (and therefore no knowledge of the numbers in each Euro class). Given the large range of emissions by Euro status, I don't think this statement is particularly defensible. Furthermore, all the data shown in Table 1/2 I think have overlapping 95% CI when making this comparison. Also, the statement "which may be attributed to the more stringent domestic goals regarding pollution, clean air, greenhouse gas emissions, energy efficiency, and innovative sustainable solutions..." is vague. Is it the case that Swedish annual technical inspections are more stringent than other countries? I doubt there is evidence for that.*

**Author Response:** We agree that the lack of Euro type information of non-Swedish HDTs indeed makes the direct comparison between the emission factors of Swedish HDTs and non-Swedish HDTs of different Euro types difficult. Based on the reviewer's comments, we considered the Swedish and non-Swedish HDTs as two major categories to compare if there are significant differences between their emission factors. We believe such comparison is useful for readers given that the non-Swedish HDTs contribute to 32% of the vehicles we studied. Since the emission data are not normally distributed, statistical significance between EF distribution of total Swedish and non-Swedish HDTs is assessed with the Kolmogorov-Smirnov test. Table R1 compares the average and median emission data and lists the p-values from the Kolmogorov-Smirnov test. The p-values are calculated at the statistical 95% confidence level. Overall, compared with non-Swedish HDTs, Swedish HDTs generally have a lower median and average $EF_{NOx}$ but there are no significant differences in the EF of other pollutants.

**Author action:** Accordingly, we revised line 328 to 332 in the revised manuscript from "Compared with non-Swedish HDTs, Swedish HDTs generally have lower EFs in terms of all the pollutants (Fig. 2 and Tables 1 and 2), which may be attributed to the more stringent domestic goals regarding pollution, clean air, greenhouse gas emissions, energy efficiency, and innovative sustainable solutions (Government Offices of Sweden, 2017). One may note that the non-Swedish HDTs was not identified according to Euro class and could contain a larger share of non-Euro VI trucks." to:

"Compared with the fleet of non-Swedish HDTs, the Swedish HDT fleet generally have a lower median and average $EF_{NOx}$ but there are no significant differences in the EF of other pollutants (Fig. 2 and Tables 1 and 2). The differences in $EF_{NOx}$ are significant at the statistical 95% CI using the Kolmogorov-Smirnov test, used in favour to typical student t-test to account for non-normality of the EF distributions. As information of Euro class, engine types and treatment technologies of non-Swedish HDTs is not available, we cannot further explore why there is a difference between the two fleets."

**Table R1.** Comparison of the average and median emission data for PM, PN, BC, NO$_x$, CO and HC from the Swedish and non-Swedish HDTs and p-values from Kolmogorov-Smirnov test.

| Pollutant | Average EF | | Median EF | | $p$ |
|---|---|---|---|---|---|
| | Swedish | Non-Swedish | Swedish | Non-Swedish | |
| PM | 95.9 mg (kg fuel)$^{-1}$ | 117.3 mg (kg fuel)$^{-1}$ | 8.1 mg (kg fuel)$^{-1}$ | 16.4 mg (kg fuel)$^{-1}$ | 0.17 |
| PN | $9.6\times10^{14}$ # (kg fuel)$^{-1}$ | $11.1\times10^{14}$ # (kg fuel)$^{-1}$ | $1.7\times10^{14}$ # (kg fuel)$^{-1}$ | $1.7\times10^{14}$ # (kg fuel)$^{-1}$ | 0.95 |
| BC | 110.9 mg (kg fuel)$^{-1}$ | 150.0 mg (kg fuel)$^{-1}$ | 2.4 mg (kg fuel)$^{-1}$ | 3.2 mg (kg fuel)$^{-1}$ | 0.25 |
| NO$_x$ | 10.7 g (kg fuel)$^{-1}$ | 13.0 g (kg fuel)$^{-1}$ | 2.7 g (kg fuel)$^{-1}$ | 6.3 g g (kg fuel)$^{-1}$ | 0.01 |
| CO | 18.6 g (kg fuel)$^{-1}$ | 19.1 g (kg fuel)$^{-1}$ | 14.5 g (kg fuel)$^{-1}$ | 13.4 g (kg fuel)$^{-1}$ | 0.98 |
| HC | 0.9 g (kg fuel)$^{-1}$ | 0.9 g (kg fuel)$^{-1}$ | 0 g (kg fuel)$^{-1}$ | 0 g (kg fuel)$^{-1}$ | 0.57 |

*2. How sure can the authors be that certain Euro classes of vehicles have certain technologies fitted? For example, is it known that any of the tested vehicles were retrofitted in some way?*

**Author Response:** We do not have specific information on after-treatment systems for each HDT in this study. We referred to the International Council for Clean Transportation (ICCT) which reported that Particulate Filters (DPFs) are required to comply with PM and PN for Euro VI HDTs (Williams and Minjares, 2016). No information about potential retrofits of tested HDTs was available for the vehicles measured in this study.

*3. Where comments are made about decreases (or changes in general) I think it is important to provide the corresponding uncertainty. It seems that in many cases that there will be overlapping confidence intervals and therefore important to convey where differences are statistically clear or not.*

**Author Response:** Thanks for your suggestions. We have conducted the Jonckheere-Terpstra test to determine if there is a statistically significant trend of pollutant emission factors based on the different Euro classes.

The null hypothesis for the Jonckheere-Terpstra test is that the distribution of pollutant emission factors is the same across the categories of Euro classes and the alternative hypothesis is that pollutant median EF decreases with more stringent Euro standards. The p-values were calculated at the statistical 95% confidence interval. The Jonckheere-Terpstra test for ordered alternatives shows that there were statistically significant trends of lower median EF$_{PM}$, EF$_{PN}$, EF$_{BC}$, EF$_{NOx}$, and EF$_{CO}$ with more stringent Euro standards from Euro III to Euro VI HDTs. However, no significant decreasing trend was evident for EF$_{HC}$ from Euro III to Euro VI HDTs. These test results are consistent with the statements we made in the main text.

**Author action:** We have added the following additions to the previous sentences.

Line 195-197: Added two sentences, this section now reads "Generally, both PM and PN emissions decreased with more stringent Euro emission standards, and especially for Euro VI where larger changes in emission characteristics were evident. These decreasing trends are statistically significant at the 95% CI using the Jonckheere-Terpstra test, a nonparametric test for trends in ordered groups. In addition to PM and PN, the emission trends of BC, NO$_x$, CO and HC with respect to the level of stringency of Euro standards were statistically examined."

Line 247-249. The sentence has been revised from "The BC emissions generally showed an overall decrease when moving towards newer Euro classes, which is similar to the EF$_{PM}$ trend with the exception of Euro

IV HDTs." to "The BC emissions generally showed a decrease from Euro III to Euro VI HDTs (Jonckheere-Terpstra test, $p<0.01$), which is similar to the $EF_{PM}$ trend with the exception of Euro IV HDTs."

Line 287-288: The sentence has been revised from "Generally, both $EF_{PM}$ and $EF_{PN}$ exhibited a decreasing trend from Euro III to Euro IV and from Euro V to EEV HDTs." to "Generally, both $EF_{PM}$ and $EF_{PN}$ exhibited a decreasing trend from Euro III to Euro IV and from Euro V to EEV HDTs (Jonckheere-Terpstra test, $p<0.01$)."

Line 322-324: The sentence has been revised from "HC emission was relatively low for all HDT types, but no obvious decreasing trend was evident for $EF_{HC}$ from Euro III to Euro VI HDTs (Fig. S8d and Table 2)." to "HC emission was relatively low for all HDT types, and no obvious decreasing trend was evident for $EF_{HC}$ from Euro III to Euro VI HDTs (Jonckheere-Terpstra test, $p=0.895$) (Fig. S8d and Table 2)."

Line 427-430: The sentences have been revised from "Particle emissions of PM, BC and to a lesser extent PN exhibited substantial reductions from Euro III to Euro VI HDTs. The gaseous emissions of $NO_x$ and CO also showed significant decrease with respect to Euro class, while the HC emission was relatively low for all the HDT Euro class types." to "Particle emissions of PM, BC and to a lesser extent PN exhibited substantial reductions from Euro III to Euro VI HDTs (Jonckheere-Terpstra test, $p<0.01$). The gaseous emissions of $NO_x$ and CO also showed a significant decrease with respect to Euro class (Jonckheere-Terpstra test, $p<0.01$), while the HC emission was relatively low for all the HDT Euro class types."

*4. How was the sigma of the background component of CO2 calculated? This was not based on upwind measurements, right? Presumably the variation in background cannot be represented by a single value and it varies also. Some more details are needed. It would be helpful to have a Figure that shows a 'typical' peak being analysed showing the concentration of CO2 and all other species. This would also help demonstrate that a measurement frequency of ~1 Hz is sufficient to capture an individual vehicle plume. Moreover, a discussion on the effect of sampling rate (and potentially different sampling rates) on the extracted plume characteristics / metrics would be beneficial.*

**Author Response and action:** Please see Fig.1 in the main text where some examples of typical temporal profiles of $CO_2$ and pollutant concentrations in the plumes are given. The $CO_2$ concentration was measured by a non-dispersive infrared gas analyser operated at 1Hz. $CO_2$ concentration for each plume was integrated and subtracted by the background $CO_2$ concentration to obtain the peak areas of $CO_2$, which were used to represent different degrees of dilution during sampling. The background $CO_2$ concentration $[CO_2]_{t_1}$ of each individual $CO_2$ peak was calculated by averaging five concentration data points just before the peak start point $t_1$. Plume pollutant concentrations were integrated and normalized by the peak area of $CO_2$ concentration to calculate corresponding pollutant emission factors of individual HDTs to compensate for different dilution levels, as expressed in Eq. (1):

$$EF_{pollutant} = \frac{\int_{t_1}^{t_2}([pollutant]_t - [pollutant]_{t_1})dt}{\int_{t_1}^{t_2}([CO_2]_t - [CO_2]_{t_1})dt} \times EF_{CO_2}. \tag{1}$$

In this study, all instruments operated at a time resolution of 1s (1Hz) or faster, which is sufficiently fast to measure pollutant concentration peaks (typically 5 to 20 s in duration) as shown in Fig. 1c-e. Thus, we do not think it would be needed to discuss this further. However, we added this statement to line 148-150, from "All the instruments were operated at least at 1Hz of sampling frequency to capture rapidly changing concentrations during the passage of a HDT." to "All the instruments were at least operated at 1Hz of sampling frequency to capture rapidly changing concentrations during the passage of a HDT, which is

sufficiently fast to measure pollutant concentration peaks (typically 5 to 20 s in duration) as shown in Fig. 1c-e."

We have compared the peak shapes under the different sampling frequency conditions by averaging 1Hz data to 2s, 3s, 4s, 5s intervals respectively. Low sampling frequencies may produce distortions to the concentration peaks (Fig. R1). For a narrow peak such as $CO_2$, lower sampling frequencies can cause difficulties in identifying the peak position and shape. These may result in calculation errors in integrated peak areas.

[Figure]

**Figure 1.** (a) Sampling site at the roadside in Gothenburg, Sweden, (b) schematic of the experimental set-up. HDT (Heavy-duty truck), RSD (Remote Sensing Device), CPC (Condensation Particle Counter), EEPS (Engine Exhaust Particle Sizer Spectrometer), TD (Thermodenuder) and HR-ToF-CIMS* (High-Resolution Time-of-Flight Chemical Ionization Mass Spectrometer) and examples of signals from three passing HDTs. Concentrations of $CO_2$, PN, non-volatile PN, PM, non-volatile PM, and $NO_x$ from (c) a typical Euro III HDTs and (d) a typical Euro VI HDTs and (e) a Euro VI HDTs with low PN emission. *The data from the HR-ToF-CIMS will be presented elsewhere.

[Figure]

**Figure R1.** Concentrations of $CO_2$ and $NO_x$ from a HDT measured under different sampling frequencies

*5. Do the authors have any information about the engine type or manufacturer to understand whether that is an important factor that could explain some of the differences observed? Earlier on in the text it is stated that these types of factor can be important in determining emissions, so it would be useful to explain this. Similarly, is there any difference in the size of vehicle sampled (e.g. by engine size or kerb weight)?*

**Author Response:** We have also investigated pollutant emission trends with respect to different manufacturers. Pollutant emission factors of HDTs under the same Euro class but from different manufacturers are compared in Fig. S9. The red solid lines in Fig. S9 represent the median EFs for the different engine manufacturers.

**Author action:** A discussion section on the influence of individual vehicle manufacturers has been added to Sect. 3.3 in the main text and Fig. S9 has been added to the supplemental materials. The following has been added, starting on line 333:

"In addition to engine Euro type, pollutant emission trends were also investigated with respect to five different vehicle manufacturers (M1, M2, M3, M4, and M5). $EF_{PM}$, $EF_{PN}$, $EF_{BC}$ and $EF_{NOx}$ of HDTs under the same Euro class but from different manufacturers are compared in Fig. S9. Since EF data was not normally distributed, statistical significance is assessed with a Kruskal–Wallis test. It is a non-paramedic analogue of the one-way ANOVA test. The p-values are calculated at the statistical 95% confidence level. No significant group difference ($p>0.05$) was observed in $EF_{PM}$, $EF_{PN}$, $EF_{BC,}$ and $EF_{NOx}$ for Euro V HDTs, i.e., HDTs from five different manufacturers show comparable emission characteristics. $EF_{PM}$, $EF_{PN,}$ and $EF_{NOx}$ of Euro VI HDTs show no dependency on manufacturers, but a significant difference was observed between M2 and M5 in $EF_{BC}$ of Euro VI HDTs ($p=0.016$). (No analysis on Euro III, Euro IV, and EEV HDTs was conducted due to the limited vehicle number from each manufacturer)."

In this study, engine size and kerb weight information were not available. However, the related information that whether the HDT is equipped with a container or not can be accessed by the captured photo, but the loading information of the container (full or empty) is not known. We conducted a Kolmogorov-Smirnov

(at 0.05 significance level) test to compare the cumulative distribution of the pollutant emission factors of HDTs with and without containers. No significant difference between $EF_{PM}$, $EF_{PN}$, $EF_{BC}$ and $EF_{NOx}$ of HDTs with and without a container ($p>0.05$) was observed.

[Figure]

**Fig. S9.** (a) $EF_{PM}$, (b) $EF_{PN}$, (c) $EF_{BC}$ and (d) $EF_{NOx}$ for Euro V HDTs and (e) $EF_{PM}$, (f) $EF_{PN}$, (g) $EF_{BC}$ and (h) $EF_{NOx}$ for Euro VI HDTs with respect to manufacturers: M1, M2, M3, M4 and M5. For an individual HDT with multiple passages, an average has been calculated and the error given is the standard deviation ($1\sigma$). The red solid lines represent the median EFs for the different engine manufacturers. Kruskal–Wallis test shows no significant manufacturer difference in $EF_{PM}$, $EF_{PN}$, $EF_{BC}$ and $EF_{NOx}$ for Euro V HDTs, whereas a significant difference was observed between M2 and M5 in $EF_{BC}$ of Euro VI HDTs ($p=0.016$).

*6. Section 4 (Atmospheric implications and conclusions) does not actually consider the atmospheric implications. I think it should – and if it did – it would strengthen the paper. For example, it would be useful*

*to consider the implications for near-road exposures and consider how UFP could evolve through coagulation etc. as plumes disperse away from roads. Reducing PM mass is clearly important but if the consequence in doing so increase PN, then that could be important.*

**Author response and action:** Thanks for the review's constructive suggestion, which can help us improve the value of our study. We have added the related discussions about implications in Sect.4.

"Reducing particle mass by DPF is clearly important but the consequence in doing so removes particle surface area available for condensation and may therefore favour nucleation mode particle formation if not the precursors of these are also reduced. Furthermore, due to the absence of larger particles, the coagulation rate is decreased and produced nucleation mode particle can retain for a longer time in the atmosphere, which has a direct influence on the evaluation of near-road human exposure."

References

Williams, M., and Minjares, R.: A technical summary of Euro 6/VI vehicle emission standards, International Council for Clean Transportation (ICCT), Washington, DC, accessed July, 10, 2017, 2016.

---

## Author Comment (AC2) · 18 Dec 2019

General comments

**Author Response:** We thank the referee for the positive and insightful comments. Our point-by-point responses to the reviewer's general and specific comments are presented below. The changes to the initial manuscript text and supplement illustrations are marked in red. Any page or paragraph reference is to the original manuscript and the reviewers' original comments are in italic. The manuscript has been updated accordingly.

*This study reports roadside measurements of 556 heavy-duty trucks in Sweden. The paper uses these measurements to investigate how several pollutants of interest vary by each truck's Euro pollutant emissions standard and carries out several additional analyses including the skewness of the distribution of emitters. In addition, the study does a very extensive comparison to past relevant literature. I think the paper is a valuable contribution to the literature and should be published after considering the comments below. The comments below that I think require the most attention have to do with adding additional analysis or at least discussion on how various amounts of exhaust dilution paper impact your particle number emission factors. This includes both variability in dilution among different trucks in your measurements, and especially differences between your study and the past work used for comparison.*

*1. Line 104: Please consider adding a short statement here related to how the uphill conditions could impact results. You discuss this later in the paper, but you might want to alert the reader to this, and point to where you discuss it.*

**Author Response and action:** Thanks for your suggestion, we have added a statement at the suggested position in the text (line 104):

"Pollutant emissions from HDTs were measured at a roadside location in Gothenburg, Sweden (Fig. 1). The HDTs passed the sampling location with an average speed of 27 km h$^{-1}$ and acceleration of 0.7 km h$^{-1}$ s$^{-1}$ on a slight uphill slope (~2°). Under such uphill driving conditions, vehicles are expected to emit higher levels of pollutants than during downhill and cruise driving. This will be further examined in Sect. 3.3."

*2. Line 163: Please discuss how you chose t1 and t2 for the integration. And more importantly, how did you deal with the different plume widths for NOx versus other pollutants? How sensitive are results to chosen t1 and t2?*

**Author Response:** In previous applications of this peak integration method to characterize particle and gaseous pollutant emission factors for heavy-duty diesel trucks, $t_1$ and $t_2$ were determined by identifying inflection points to the left and right of the individual pollutant peaks (Dallmann et al., 2011; Preble et al., 2015). A similar method was used here, examples of chosen $t_1$ and $t_2$ are given in Fig. R1.

**Author action:** We have added a statement after line 166 in the revised manuscript "The time interval of $t_1$ to $t_2$ represents the period when the instruments measured the concentration of an entire pollutant peak from an individual HDT (see Fig. 1c-e). Dallmann et al. (2011) and Preble et al. (2015) used the concepts of inflection points to identify $t_1$ and $t_2$. In our study, $t_1$ and $t_2$ were determined similarly: $t_1$ is the point before the pollutant concentration intensity increases abruptly and $t_2$ is when the intensity becomes relatively flat and undistinguishable compared to background levels. It is noted that the integrated peak intensity is insensitive to the exact location of $t_2$ since the added integrated signals at or beyond this point are fluctuating around zero. $t_1$ and $t_2$ were determined independently of each pollutant peak to account for differences in the time response of individual instruments to the exhaust plume."

If we use the same $t_1$ and $t_2$ for different pollutants, the integrated peak area would be biased, so the choices of $t_1$ and $t_2$ are sensitive to the peak width, or more specifically pollutant type.

[Figure]

**Figure R1.** Concentrations of $CO_2$, PN, non-volatile PN (black line), PM, non-volatile PM (black line), and $NO_x$ from (a) a typical Euro III HDT and (b) a typical Euro VI HDT and (c) a Euro VI HDT with low PN emission.

*3. Line 188-200: See major comments above. PN measurements would be highly dependent on the amount of dilution the plume has undergone between the engine and the measurement. I would imagine this would contribute to differences between your measured emission factors and emission standards. What are the dilution requirements when certifying for Euro standards?*

*Line 200-202: Could variability in dilution contribute to the scatter too? Please think this through for all sections that discuss PN emissions results.*

**Author Response:** To compensate for different dilution levels, particle emissions from individual HDTs were normalized by the $CO_2$ concentration as illustrated in Eq. (1).

$$EF_{pollutant} = \frac{\int_{t_1}^{t_2}([pollutant]_t - [pollutant]_{t_1})dt}{\int_{t_1}^{t_2}([CO_2]_t - [CO_2]_{t_1})dt} \times EF_{CO_2} . \tag{1}$$

In principle, EFs calculated by Eq. (1) would not be dependent on the amount of dilution the plume has undergone between the engine and the measurement if there is no transformation of the pollutants. Nowadays, the legislation in Europe prescribes the Constant Volume Sampling (CVS) as the reference sampling for particle emission certification, in which the sample mixed gas of exhaust and diluent gas is controlled to have a constant flow rate. The dilution ratio has been left out of direct regulation and is only implicitly controlled by the need to achieve sufficient exhaust cooling before particle

sampling (Ntziachristos et al., 2004). The regulations regarding PN is following the PMP protocol, where only the solid particle fraction > 23 nm is accounted for, hence a fraction that is less sensitive to the dilution.

However, the effective emission of all nucleation mode particles is depending on nucleation, coagulation, and evaporation that could be happening on the time scale of the dilution and cooling of the exhaust gases. The relationship between the number emission factor of nucleation mode particles and $CO_2$ peak areas are shown in Fig. R2. Higher $CO_2$ concentrations indicated a lower dilution level. Since EF data was not normally distributed, the strength and direction of the association between nucleation mode particle number emission factors and $CO_2$ peak areas were assessed with the Kendall's tau-b correlation. It is a nonparametric alternative to the Pearson's correlation. The p-values are calculated at the 95% confidence level. Intuitively, one would expect that lowering dilution, i.e., increased observed $CO_2$ concentrations, might result in a higher $EF_{PN}$ in nucleation mode. However, as shown in Fig R2, no significant enhancement of the formation of nucleation mode particles was evident under lower dilution levels and the nucleation mode particle number emission factors remained practically constant for all Euro III-V HDTs ($p>0.05$) while Euro VI HDTs showed a weak negative correlation (correlation coefficient of −0.3, $p<0.01$), i.e. opposite to what could be expected from coagulation/evaporation. This demonstrates that any potential dilution effect on the variability of the measurements was limited and well represents typical dilution one observes at kerb-sites.

[Figure]

**Figure R2.** The relationship between number emission factor of nucleation mode particles and $CO_2$ peak areas

*4. Line 240-241: Please make sure to include text in figure captions when you are not including data from all trucks. Are you sure that leaving these data out doesn't lead to a problem with biasing the results? I would imagine that if you are not including results for trucks that have measured concentrations below measurement detection limits, you'd be leaving out the cleanest trucks (though could also be due to the plume missing the sample line). Please think this through for all sections that report results that remove trucks with measurements below detection limits.*

**Author Response:** Thanks for the suggestion. We have added the description of the excluded data in the figure captions. We apologize for the confusion caused; we indeed included the emission data lower than

the set detection limit (four times the standard deviation of the pollutant background signal) into the analysis throughout the whole manuscript. As we mentioned in Sect. 2.3, emission factors for plumes with pollutant concentrations lower than our set detection limit were replaced by the minimum value among all recorded emission factors ($EF_{min}$) rather than being omitted to avoid inflating emissions from low-emitting HDTs. The concentration data below the detection limits were removed from the figure only for the purpose of a clearer figure presentation.

**Author action:** We have added the following statement after line 253-254 in the revised manuscript:

"HDTs with either $EF_{NO2}$ or $EF_{NOx}$ lower than the detection limits of the instruments were removed in Fig. 2d for illustration purposes, while all the presented statistical analyse include all the data as outlined above."

5. Table 1: I don't understand how you've categorized this table. For example, I see studies in this table that are not performed in Europe but are under the Euro VI category. Also, I noticed papers that you are citing in the study and that have emission factor results, but are not in this table. Please ensure you have considered all relevant studies.

**Author Response action:** Thanks for review's suggestions. We have rearranged some rows in Table 1 and Table 2 and separated the studies of non-European HDV emissions with Euro VI HDV emissions in each Table. Additionally, more emission factor results have been added to the tables.

[revised manuscript text omitted]

*6. Figure 2: For Euro III, it seems that the EF for black carbon is higher than for PM. How could this be?*

**Author Response:** In the conversion of number concentrations to particle mass, particle sphericity and unit density were assumed. This might lead to an underestimation of PM. Nevertheless, we use this in the paper for easy comparison with the literature using similar experimental methodologies (Hallquist et al., 2013; Liu et al., 2019; Preble et al., 2015; Watne et al., 2018) in which unity density was used for the calculation of $EF_{PM}$.

In Fig. 2a, $EF_{PM}$ includes both semi-volatile and non-volatile fractions. Using measurements with and without thermal desorption, we found that the ratio of $EF_{non\text{-}volatile\ PM}$ to $EF_{PM}$ is generally higher for Euro III HDTs than other Euro type HDTs (Fig. R3). This explains why only Euro III HDTs show a higher $EF_{BC}$ than $EF_{PM}$. In fact, there is a good linear relationship at $EF_{PM}$ larger than 1 mg (kg fuel)$^{-1}$ between the BC mass measured by the Aethalometer and the non-volatile particle mass measured at the outflow of a TD by the EEPS (Fig. S3). Compared to the EEPS, the detection limit of the Aethalometer is five times higher, which may influence the correlation between BC and PM at low mass loading conditions (when $EF_{PM}$ is lower than 1 mg (kg fuel)$^{-1}$).

[Figure]

**Figure R3.** $EF_{non-volatile PM}$ / $EF_{PM}$ for Euro III to Euro VI HDTs. Non-detectable pollutant emission signals for captured plumes have been replaced by $EF_{min}$. For box-and-whisker plots, the top and the bottom line of the box are 75[th] and 25[th] percentiles of the data, the red line inside the box is the median, and the top and bottom whiskers are 90[th] and 10[th] percentiles.

[Figure]

**Fig. S3.** Relationship between $EF_{BC}$ measured by the Aethalometer and $EF_{non-volatile PM}$ measured by the EEPS in the TD line (unity density of particles was assumed).

*7. Figure 4: This is very interesting. You might consider comparing these size resolved emission factors to previous studies that report similar EFs.*

**Author response and action:** Thanks for your suggestion, we have added the following discussion in red and Fig. 4f to the manuscript:

The $EF_{PN}$ of the accumulation mode particles shows a decreasing trend from Euro III to EEV HDTs. The accumulation mode of the Euro VI HDTs was insignificant. For heavy-duty diesel engines without a particulate filter, nucleation mode particles are mainly formed from organics. For vehicles with DPF both

organics and the fuel sulphur content might influence the formation of nucleation mode particles (Vaaraslahti et al., 2004). Thiruvengadam et al. (2012) found a direct relationship between exhaust nanoparticles in the nucleation mode and the exhaust temperature of the DPF-SCR equipped diesel engine. These factors lead to high variability in the nucleation mode fraction of $EF_{PN}$. Figure 4f shows that HDVs with DPF (dashed lines) exhibited lower emissions of accumulation mode particles, with no significant reduction in nucleation mode particles when compared to HDVs without DPF (solid lines). In general, the absence of significant accumulation mode particles from Euro VI HDTs was consistent with observations made from DPF equipped HDVs. High emissions of accumulation mode particles from Euro III HDTs were consistent with measurements from HDVs without DPF in previous studies (Liu et al., 2019; Hallquist et al., 2009; Preble et al., 2015).

[Figure]

Figure 4f. Comparisons of mean size-resolved $EF_{PN}$ of HDVs in this study and previous studies.